# R-loop induced G-quadruplex in non-template promotes transcription by successive R-loop formation

Chun-Ying Lee[1], Christina McNerney [2], Kevin Ma[1], Walter Zhao [3], Ashley Wang [1] & Sua Myong [1,4✉]

G-quadruplex (G4) is a noncanonical secondary structure of DNA or RNA which can enhance or repress gene expression, yet the underlying molecular mechanism remains uncertain. Here we show that when positioned downstream of transcription start site, the orientation of potential G4 forming sequence (PQS), but not the sequence alters transcriptional output. Ensemble in vitro transcription assays indicate that PQS in the non-template increases mRNA production rate and yield. Using sequential single molecule detection stages, we demonstrate that while binding and initiation of T7 RNA polymerase is unchanged, the efficiency of elongation and the final mRNA output is higher when PQS is in the non-template. Strikingly, the enhanced elongation arises from the transcription-induced R-loop formation, which in turn generates G4 structure in the non-template. The G4 stabilized R-loop leads to increased transcription by a mechanism involving successive rounds of R-loop formation.

[1] Department of Biophysics, Johns Hopkins University, Baltimore, Maryland 21218, USA. [2] Department of Biology, Johns Hopkins University, Baltimore, Maryland 21218, USA. [3] Department of Biomedical Engineering, Johns Hopkins University, Baltimore, Maryland 21218, USA. [4] Physics Frontier Center (Center for Physics of Living Cells), University of Illinois, Urbana, Illinois 61801, USA. ✉email: smyong@jhu.edu

G-quadruplex (G4) is a noncanonical nucleic acid structure formed by guanine-rich sequences both in vitro and in vivo[1]. When surveyed at the genome scale, these potential G4-forming sequences (PQS) are found unevenly distributed across the human genome. Its specific enrichment in the promoter region of regulatory genes, splice junctions, and the 5′-untranslated region (5′-UTR) of highly transcribed genes strongly argues for a potential role of PQS in gene regulation[2–5]. For example, it has been demonstrated that PQS located at 5′-UTR can regulate transcription and translation levels in vivo[6–8]. Moreover, the biological role of PQS is also implicated in the disease context, where mutation and loss-of-activity of certain G4-resolving helicases (e.g., BLM, WRN, and XPB/XPD) lead to an altered pattern of G4 formation and transcription factor (TF) binding in the promoter regions[9]. In addition, PQS is also found in the promoter of many potent oncogenes, where it is proposed to regulate the expression of these genes[10]. However, it is mechanistically not clear how PQS regulates the processes of transcription and translation.

The effect of transcriptional regulation by PQS depends on several factors such as the composition of PQS sequence, its location with respect to transcription start sites (TSSs), and the orientation (template (T) or non-template (NT))[11]. PQS in the promoter has been shown to regulate gene expression in the presence of G4-binding ligand, which stabilizes G4 structure[12–15]. Recently, orientation of PQS was shown to cause up- or down-regulation of transcription by inducing G4 formation via oxidative DNA damage repair pathway[16–18]. In Bloom and Werner Syndromes, the 5′-UTR PQS on the T and NT strands may suppress and enhance gene expression, respectively[19]. The repression of transcription by T-PQS may be due to the formation of G4 that can arrest RNA polymerase (RNAP)[20]. However, it remains elusive how PQS on the NT influences transcription. Furthermore, although previous studies have shown correlation between PQS at 5′-UTR and transcription level, there is no direct evidence to show whether G4 actually forms during transcription reaction.

Another structure that may arise from transcribing the 5′-UTR PQS-rich regions is the R-loop in which nascent RNA invades into upstream double-stranded (ds) DNA and forms a DNA: RNA hybrid with the T strand, displacing the NT strand[21]. This R-loop structure was previously observed by an electron microscopy study when PQS-rich region in plasmid undergoes transcription[22]. Subsequent studies using electrophoretic mobility shift assays (EMSAs) reported a potentially coupled G4 and R-loop formation during transcription[23–25]. Previous work also addressed the role of R-loop mediated by G-rich sequences on transcription level[26,27]. In addition, it has been reported that G/C content is coupled to R-loop formation and transcriptional pausing through human genome by R-Chip Rnase H probing method[28]. Furthermore, G4 stabilizing ligands induced R-loop formation in cells, which in turn led to genome instability[29]. Taken together, these previous results suggest a possibility that both G4 and R-loop may be induced by transcription, yet the formation of G4 and R-loop, and their effect on transcription have not been directly addressed.

We sought to investigate the sequence and orientation effect of 5′-UTR PQS in transcription by applying both ensemble and single-molecule methods. To gain mechanistic understanding, we devised sequential single-molecule platforms to dissect each stages of transcription. Our results reveal that PQS orientation rather than its sequence impacts transcription significantly and PQS on NT strand enhances transcription. The increased transcription through PQS in NT arises from the prominent R-loop formation stabilized by G4 formed in NT. Remarkably, the R-loop leads to successive cycles of R-loop formation, giving rise to transcriptional activation. In summary, we provide a comprehensive mechanism by which PQS impacts transcription by forming R-loop and G4 structures.

## Results

**PQS orientation regulates RNA production rate**. To examine the effect of PQS in transcription, we set up an ensemble in vitro transcription assay in which a reporter probe (pre-quenched molecular beacon) fluoresces upon binding the transcribed RNA. We prepared a DNA substrate (840 bp) containing T7 promoter and PQS positioned downstream of TSS (+41) (Fig. 1a). The reporter probe is an 18 nucleotide (nt) single-stranded (ss) DNA bearing a sequence complementary to the transcribed RNA and end-modified with Cy3 and black hole quencher (BHQ2) such that the quenched (dark) probe fluoresces upon annealing to the mRNA product (Fig. 1a). First, the mRNA product (800 nt) was checked by running the in vitro transcription product on a 3% agarose gel (Fig. 1b). The quantification of gel bands revealed that the rate of transcription was higher (~30%) for the PQS in NT (PQS-NT) (Supplementary Table 2) as compared with the template (PQS-T) (Fig. 1c).

Next, we used a plate reader to measure the real-time transcription reaction for different PQS constructs with varying G4-forming potential sequences mapped previously by NMM (N-methyl mesoporphyrin IX) fluorescence[30,31] (Fig. 1d and Supplementary Fig. 1). The linear slope of the fluorescence increase for each PQS construct (by using the same reporter probe) was normalized against the non-PQS control (Supplementary Table 2) and plotted in the order of the GQ folding propensity (Supplementary Fig. 1). The result shows that the orientation rather than sequence composition of PQS influences transcription, and the PQS-NT promotes while the same set of PQS-T diminishes the mRNA production compared with the non-PQS control (Fig. 1e). We note that the reporter probe concentration was limited for obtaining reproducible transcription readout without increasing too much background. Therefore, the differences in transcriptional output may be underestimated compared with the other results based on EMSA and single-molecule assays presented below.

**mRNA production is promoted with PQS in the non-template**. Next, we tested the PQS orientation effect by single-molecule platform using the same reporter probe. We chose one representative PQS sequence (cMyc) to conduct a single-molecule study and examine orientation dependence. The DNA template was tethered to single-molecule surface by biotin-NeutrAvidin linkage and the same RNAP reaction mix including the reporter probe was applied to the surface. As expected, we observed an appearance of single Cy3 fluorescence spots, indicating the mRNA production from individual DNA template probed by the subsequent annealing with the reporter probe (Fig. 1f). The dwell time of spikes was used to distinguish the real signal (>500 ms) from nonspecific probe binding (<100 ms). The Cy3 spikes were accompanied by a gradual increase in the Cy3 background, likely resulting from the accumulation of freely diffusing mRNA-reporter complex in the detection chamber. Indeed, the slope of the Cy3 background increase was steeper in higher NTP concentration, reflecting a higher rate of mRNA production. The cumulative counts of Cy3 burst (>150 traces, 6 min) (Fig. 1g) displayed higher rate of increase (~30%) for PQS-NT. Furthermore, the dwell time (spike-to-spike) analysis also revealed higher frequency (~30 ~ 50%) in PQS-NT than in PQS-T (Fig. 1h), confirming the correlation between the single-molecule and bulk assay.

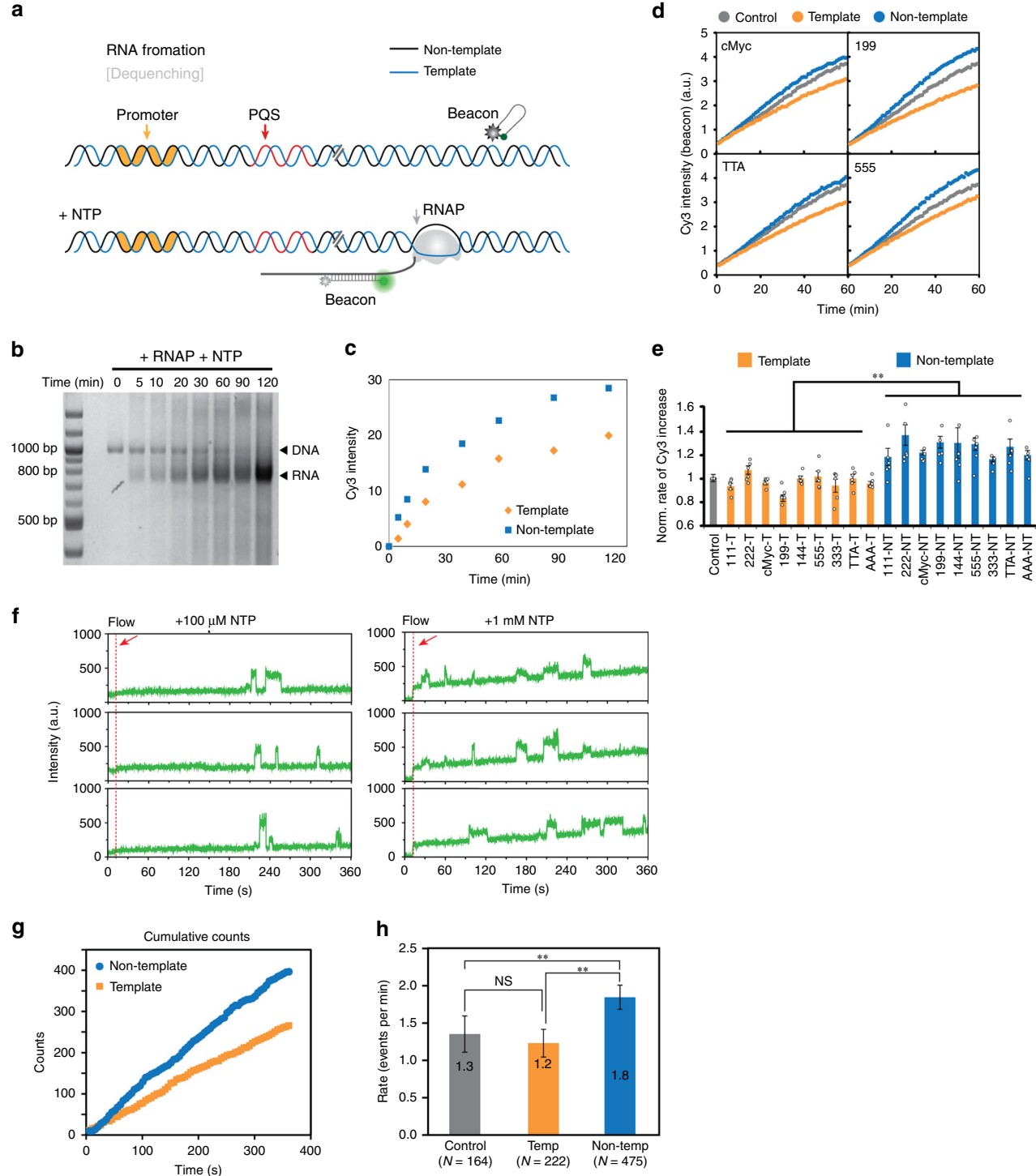

**Fig. 1 PQS orientation effect on RNA production detected by molecular beacon in both ensemble and single-molecule transcription assay. a** The schematic of beacon assay. Promoter, PQS, and RNAP are colored in orange, red, and gray, respectively. Transcribed RNA is annealed with beacon, dequenching the Cy3 signal. **b** Transcribed RNA mixed with beacon is distinguished from DNA substrate by 3% agarose gel electrophoresis. Shown is a representative result from two independent experiments of one substrate. **c** Quantification of Cy3 intensities from **b**. **d** Real-time Cy3 signal measurements of individual PQS samples by plate reader. **e** Initial RNA production rate is calculated from the early linear part of curve in **d**. Data are presented as mean ± SEM of $n = 6$ independent experiments. The exact data and $P$-values are provided in Supplementary Table 3.1. **$P < 0.005$ (two-sided unpaired $t$-test). Shown in **e** only represents the significance between template and non-teamplate. **f** Single-molecule assay detects Cy3 signal bursting, while annealing with transcribed RNA close to surface. Red arrow indicates the initiation point when RNAP, NTP, and beacon mix is flown into detection chamber. **g** Cumulative counts of Cy3 burst from more than 150 traces (as shown in **f**). **h** The frequency of Cy3 bursts calculated from 1 mM NTP measurement (as shown in **f**). Data are presented as values ± SEM. $N$ value indicates the number of independent measurements. Statistics is described in "Methods" and Supplementary Table 3.2, and the data are provided in Source Data file. **$P < 0.01$, NS: nonsignificant (one-sided Kruskal–Wallis test).

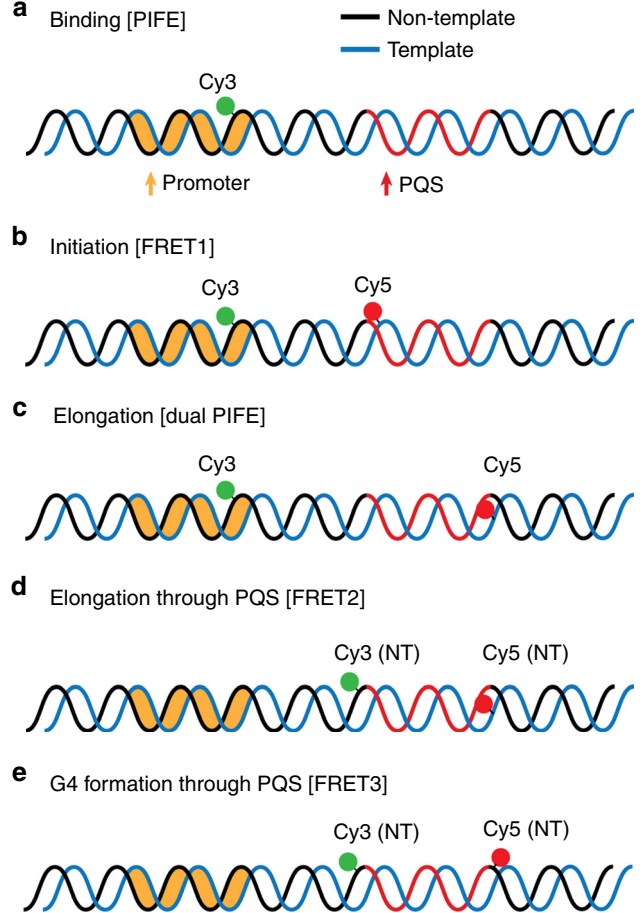

**Fig. 2 DNA constructs for single-molecule assay. a** smPIFE construct is labeled Cy3 at promoter. **b** smFRET1 construct with both dyes upstream of PQS. **c** Dual-PIFE construct is labeled Cy3 at promoter and Cy5 downstream of PQS. **d** smFRET2 construct with Cy3 and Cy5 in the non-template and template strand, respectively. **e** smFRET3 construct with both dyes in the non-template strand. Template, non-template strand, and promoter are colored in blue, black, and orange, respectively. PQS is shown as red in non-template, for example.

**Single-molecule detection platforms.** To define the molecular mechanism responsible for the enhanced transcription in PQS-NT, we devised a series of single-molecule assays to dissect the effect of PQS at each stage of transcription including promoter binding, initiation, elongation, progression through PQS and the formation of quadruplex structure (Fig. 2). To probe each step, we took advantage of several single-molecule detection platforms including single- and dual-color single-molecule Protein Induced Fluorescence Enhancement (smPIFE)[32,33] and single-molecule Förster Resonance Energy Transfer (smFRET)[34,35]. Briefly, smPIFE enables detection of protein binding near the fluorophore, resulting in two- to threefold increase in the fluorescence signal[36]. We monitored RNAP binding by one-color smPIFE (Fig. 2a). The movement of RNAP from initiation to PQS element was monitored by dual-color smPIFE (Fig. 2c). The smFRET that measures the distance change between the two FRET pair dyes, Cy3 and Cy5, was used to detect the structural changes induced by the opening of transcription bubbles[37] at two different positions along the DNA template (Fig. 2b, d, FRET1 and FRET2). In addition, another smFRET construct reported on the G4 formation (Fig. 2e, FRET3).

**PQS orientation has no effect on RNAP binding and initiation.** Binding of RNAP was detected via Cy3 PIFE signal. The Cy3

located at the promoter (-4 position) showed an increased signal upon RNAP binding at the promoter (Fig. 3a). The pulses of Cy3 intensity was only observed in the presence of RNAP (Fig. 3b) and the linear correlation between the PIFE frequency and the applied RNAP concentration confirmed the PIFE signal as a valid proxy for RNAP binding (Fig. 3c). Based on this result, we set 100 nM RNAP as an optimal concentration suitable for detecting frequent, yet well-separated binding events. When tested for non-PQS control and PQS-NT vs. PQS-T, we observed no significant difference, indicating PQS has no effect on RNAP binding (Fig. 3d).

Initiation of transcription induces transcription bubble which increase FRET signal in our assay (Fig. 3e). We adopted the previously established smFRET assay by placing Cy3 and Cy5 at −4 and +17 positions, respectively (0.4 FRET, FRET1)[37]. The RNAP binding without NTP produces PIFE signal in both dyes (Supplementary Fig. 2b). In the presence of NTP, we observe a series of FRET spikes to 0.7, indicating the opening of the transcription bubble (Fig. 3f, FRET1). As expected, the frequency of initiation increased as a function of NTP concentration (Fig. 3g). We also observed that the frequency of RNAP binding-only traces (PIFE without FRET) was substantially diminished in high NTP concentrations for all three constructs (Fig. 3h). Furthermore, the abortive synthesis which produces a characteristic FRET fluctuation[37] (Supplementary Fig. 2c) constituted about 20% of initiation events for all three constructs, not showing significant difference (Supplementary Fig. 2d). Taken together, both binding (PIFE) and initiation (FRET1) assay verify that the PQS does not influence either the binding or initiation stage.

**PQS in the non-template leads to more successful elongation.** We next employed a "Dual-PIFE" assay to track the RNAP movement from the promoter to the end of PQS site (Fig. 4a). The Cy3 and Cy5 positioned at −4 and +35, respectively (Fig. 4a), are nearly 40 base pairs (~13 nm) apart, which put them outside of the FRET-sensitive range (3–8 nm) but allows for two independent PIFE detections by simultaneous two-color excitation (Fig. 4a). Although addition of RNAP without NTP exhibited burst of green PIFE signals only, applying RNAP with NTP induced Cy3 PIFE followed by Cy5 PIFE in succession (Fig. 4b), reporting on the expected movement of RNAP from the promoter to the PQS site. The Dual-PIFE assay produced either the successful elongation signal denoted by successive Cy3–Cy5 PIFE or unsuccessful elongation displayed by only Cy3 PIFE without Cy5 PIFE (Fig. 4c). We note that the lack of Cy5 PIFE is not due to photobleaching based on the dual excitation scheme that enables continuous detection of Cy5 signal. Approximately 52% successful elongation was displayed in non-GQ control, whereas 44% and 58% was observed for PQS-T and PQS-NT, respectively (Fig. 4d), suggesting the higher rate of successful elongation in PQS-NT than the non-PQS and the PQS-T.

**Elongation through PQS is increased in PQS-NT.** Next, we sought to examine the elongation through PQS by FRET2 construct in which Cy3 and Cy5 are located directly adjacent to PQS, resulting in 0.3 FRET (Fig. 5a, FRET2). Only when both RNAP and NTP were added together, a periodic signal spikes were observed, and each event consisting of a short-lived Cy3 PIFE followed by FRET increase (Fig. 5b, bottom), which appeared in PQS-NT, non-PQS, and PQS-T constructs (Supplementary Fig. 3). The frequency of PQS elongation events analyzed by taking dwell times ($\tau_{off}$) showed a higher frequency at PQS-NT than PQS-T, in agreement with our previous results (Fig. 5c). Unexpectedly, we noticed that the FRET histogram displayed a

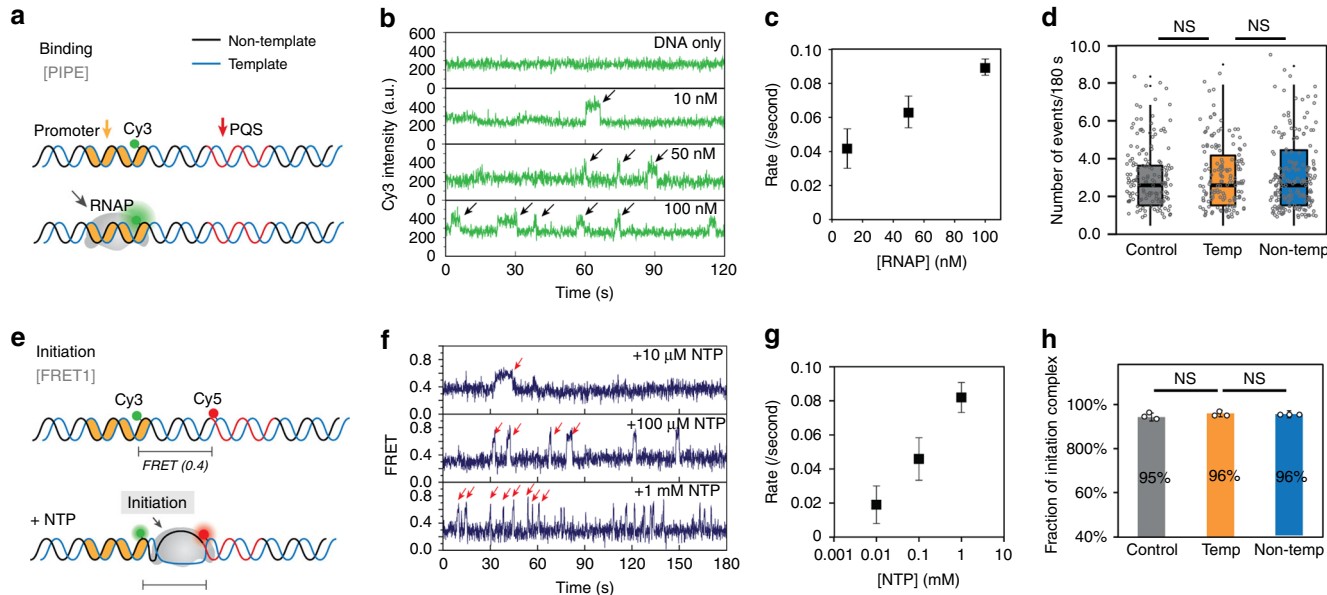

**Fig. 3 PQS orientation has no effect on binding and initiation. a** Single-molecule PIFE assay to study the RNAP binding. **b** As shown in **a**, the binding of RNAP to promoter induces Cy3 PIFE signal indicated by a black arrow. Each trace represents one example of protein titration from 0, 10, 50, and 100 nM. **c** The RNAP binding rate is calculated from the frequencies of Cy3 PIFE signal. **d** The number of events in one trace (3 min) averaged from more than 150 traces. **e** Single-molecule FRET assay to study transcription initiation. Opening the transcription bubble (gray arrow) causes a jump transition of FRET from 0.4 to 0.7. **f** As shown in **e**, the initiation event can be detected as a FRET burst (red arrow). Each trace represents one example of NTP titration from 10 μM, 100 μM, and 1 mM. **g** The initiation rate is calculated by the frequencies of FRET bursts. **h** Fraction of initiation is calculated from the population of binding and initiation in the total events. For **c**, **g**, data are presented as mean ± SEM, obtained as described in "Methods." Numbers of independent measurements are from 40 to 479. Exact mean values are provided in Supplementary Tables 3.3 and 3.5. Raw data points are provided as a Source Data file. For **d**, numbers of counts from individual traces were plotted; the bottom and top of the box present the first and third quartile, respectively; the band inside the box shows the mean and the whiskers show the upper and lower extremes. Statistical analysis performed by two-sided unpaired *t*-test shows no significance (NS). Exact *P*-values are provided in Supplementary Table 3.4; raw data points are provided as a Source Data file. For **h**, data are presented as mean ± SEM of $n = 3$ independent experiments. Statistics analysis performed by one-sided Pearson's $\chi^2$-test shows no significance (NS). Exact *P*-values are provided in Supplementary Table 3.6. How to distinguish binding and initiation is described in Supplementary Fig. 2.

prominent 0.1 FRET peak, especially in the PQS-NT in addition to the 0.3 FRET peak (from DNA alone) (Fig. 5d). What gives rise to the 0.1 FRET state?

**PQS-NT induces R-loop that increases transcriptional output**. We tested whether the 0.1 FRET species can be separated from the 0.3 duplex DNA on EMSA by running reaction mixtures collected at 10 min intervals for 1 h. The samples were treated with 0.1% SDS to denature proteins and eliminate RNAP–DNA complex. The result on PQS-NT showed the expected 90 bp band and an additional up-shifted band (migrating at 500 bp), which becomes more intense as a function of transcription time (Fig. 5f, left side). One possible structure that forms in this context is an R-loop, a transcription intermediate that arises from annealing between transcript and T, whereas the NT strand remains displaced[21,38–42]. The RNAP can induce a local negative super-coiling, which may promote R-loop formation[43–47]. In addition, guanine-rich sequence displays higher potential for R-loop formation due to the higher thermal stability of rG/dC base pairing[48]. The RNase H treatment, which selectively digests the RNA strand of the R-loop, completely removed the up-shifted bands on the EMSA gel, confirming the R-loop formation by PQS-NT. (Fig. 5f, right side). Furthermore, the RNase H digestion led to immediate disappearance of the 0.1 FRET peak, indicating the 0.1 FRET as the R-looped state (Fig. 5e). Taken together, the R-loop is formed and accumulated during transcription with the highest propensity in PQS-NT, followed by non-PQS control and the PQS-T (Supplementary Fig. 4a–c). The extent of R-loop formation was correlated with RNAP concentration with the most

prominent impact in the PQS-NT (Fig. 5h, i). Furthermore, the fraction of the up-shifted DNA corresponding to R-loop on EMSA gel showed a linear correlation with the fraction of 0.1 FRET peak in the smFRET assay (Fig. 5g and Supplementary Fig. 4d, e), confirming that they both represent the R-loop. Therefore, our results indicate that the enhanced transcription of PQS-NT arises from the higher R-loop formation than in control and PQS-T.

**Transcription is enhanced by successive R-loop formation**. How can R-loop promote transcription if the R-loop were to act as a physical barrier to RNAP? The continuous production of transcript concomitant with the progression of the R-loop formation suggests that R-loop is a transcriptionally active state (Fig. 5f, green bands as RNA and Supplementary Fig. 7 left side). Based on this observation, we hypothesized that there is a successive R-loop formation during transcription i.e. an R-loop formed in the first round of transcription is replaced by another R-loop formed in the second round and so on, leading to an increased mRNA output (Fig. 6a). In smFRET, we detect distinct moments of FRET transition from 0.3 (duplex DNA) to 0.1 state (R-loop), followed by frequent peaks of green PIFE at 0.1 FRET, likely indicating the continuing transcription events mediated by successive R-loop formation (Fig. 6b). We note that the 0.1 FRET state is not due to photobleaching of the Cy5 dye as confirmed by a direct excitation of Cy5 signal by the red laser at the end of each movie (red spike at 175 s, Fig. 6b).

To directly test the R-loop replacement model, we performed a pulse-chase experiment in which pulse of Cy5-labeled UTP

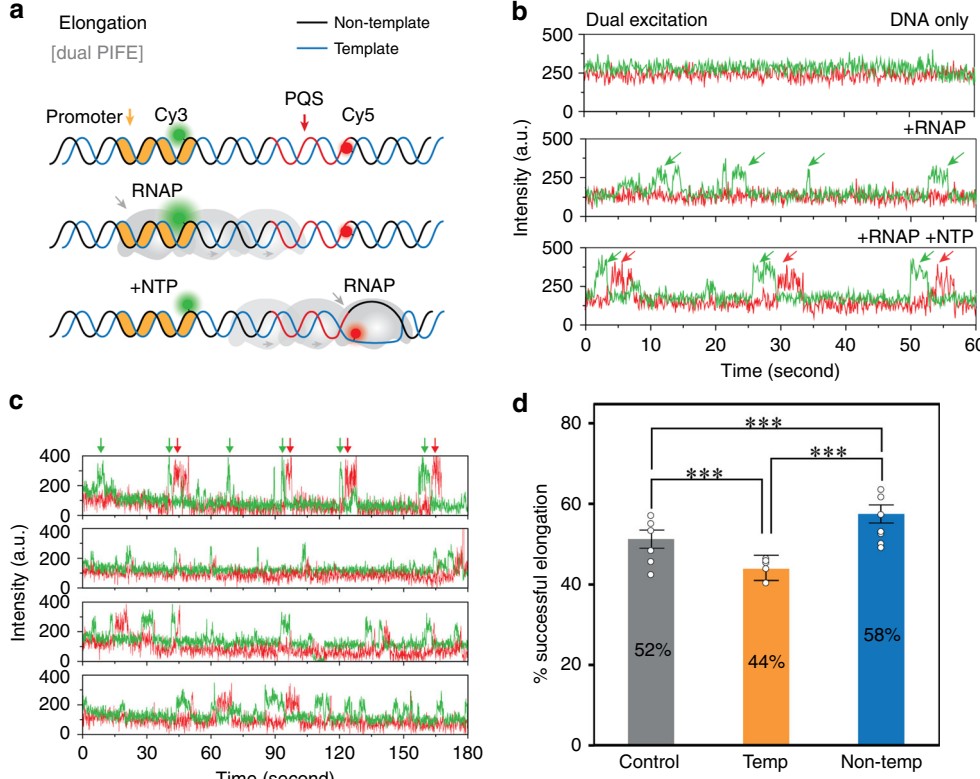

**Fig. 4 PQS orientation mediates elongation events by dual-PIFE assay. a** Experimental scheme of dual-PIFE assay. The two dyes are separated out of FRET-sensitive distance. Dual excitation of 532 nm (green) and 641 nm (red) lasers are applied simultaneously. **b** Trace examples demonstrate the events of single PIFE (green only) and dual-PIFE (green and red). **c** Examples of both single- (green) and dual-PIFE signal (red) that can exist in the same trace, indicating elongation is not always successful. **d** The fraction of successful elongation event is calculated by counting the number of dual-PIFE events out of total events. Data are presented as mean ± SEM of $n = 7$ independent experiments. Statistical significance, ***$P < 0.001$ was performed by one-sided Pearson's $\chi^2$-test. Exact mean and $P$-value are provided in Supplementary Table 3.7.

(mixed with unlabeled ATP, CTP, and GTP), was chased by an unlabeled NTP mix. The Cy5 intensity in R-loop monitored by EMSA as a function of transcription time increased upon the pulse and decreases after the chase, indicating the incorporation of Cy5-UTP into R-loop and subsequent replacement by unlabeled NTP, while in the R-looped state, supporting the model (Fig. 6c). Furthermore, we applied the same sample on single-molecule platform, where R-looped DNA (formed with Cy5-UTP) was immobilized on a slide surface. DNA and R-loop RNA were visualized by Cy3 and Cy5 signal, respectively. RNAP without NTP was flown as a negative control and the constant Cy5 count indicated R-loop RNA is stable on DNA before replacement. Upon adding RNAP and unlabeled NTP mix to initiate the RNAP reaction, we observed a decrease of Cy5, confirming the R-loop replacement due to active transcription (Fig. 6d). We note that the lower rate of R-loop replacement than the rate expected from the Cy3 spikes seen in FRET2 construct (Figs. 5c and 6b) is likely due to the dye-labeled UTP in the transcript, which cannot be removed by RNAP as efficiently.

**R-loop leads to G4 formation in NT during transcription.** One critical question we have not addressed yet is regarding the conformation of the NT strand, especially in the context of R-looped state. Based on its PQS sequence, G4 can potentially form in NT. In fact, when the PQS-NT was treated with RNase H, there was an appearance of a high FRET peak (0.9, 20~30%), which did not show up in control and PQS-T (Fig. 5e left). Such partitioning pattern indicates that the 0.1 FRET state may include two different R-looped states, i.e., R-loop with and without the G4

formed on NT. To test for G4 formation more directly, we designed smFRET3 construct in which both dyes were located on NT strand at either side of PQS (Fig. 7a). To establish a calibration, we measured FRET for the linear duplex and pre-formed G4 DNA in which the G-rich strand is folded separately in the presence of 40% PEG and 100 mM KCl prior to annealing with the C-rich strand[31]. The linear duplex and pre-formed G4 DNA displayed one sharp FRET peak at 0.25 and 0.9, respectively (Fig. 7b).

Strikingly, the FRET histogram collected after 10 min of transcription showed a clear 0.9 FRET peak, strongly indicating the G4 formation on NT. An additional 0.4 FRET peak formed along with 0.9 FRET and increased over time. Therefore, the 0.4 and 0.9 represent the two populations of R-loop states including R-loop without G4 (0.4 peak) and R-loop with G4 (0.9 peak), which could not be distinguished by the previous FRET2 construct (0.1 peak, Fig. 5d). When digested with RNase H, 0.4 peak completely disappeared, whereas 0.9 FRET peak remained at 30% (Fig. 7c), which matches the high FRET peak (0.9) in FRET2-NT (Fig. 5e), both corresponding to the G4 formed on NT strand. In summary, the FRET3 peaks partition to linear duplex (0.25, 30%), R-loop (0.4, ~40%) and G4/R-loop (0.9, ~30%) in FRET3-NT (Fig. 7c), whereas the FRET2 peaks divided into linear duplex (0.3, 30%) and both R-loops (0.1, 70%) (Fig. 5g NT). Consequently, both constructs reported on the same fraction of R-loop formation, but the previous 0.1 FRET peak actually contained two components, which can be distinguished by FRET3. As a control, we prepared a FRET3 construct containing a non-PQS sequence, which only showed two peaks,

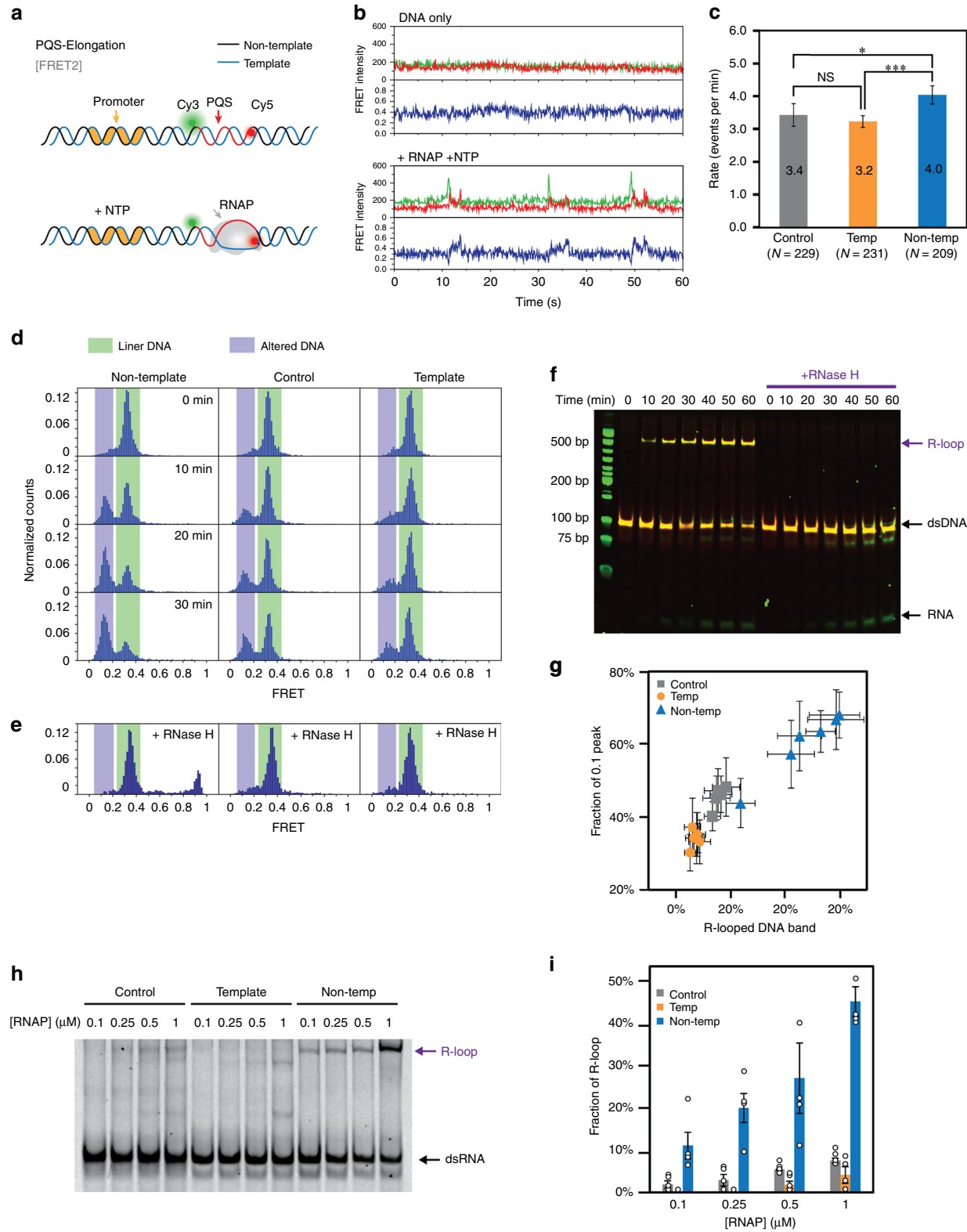

0.25 and 0.4, but no peak at 0.9 consistent with no G4 formation. Only the 0.4 peak was removed by RNase H treatment, confirming that 0.4 correspond to R-loop in both constructs and the 0.9 peak represents G4 folding only seen in PQS-NT.

Furthermore, the smFRET3 (PQS-NT) trajectories reveal several mechanistic details (Fig. 7d). First, the dynamic exchange between 0.25 and 0.4 FRET states indicates that the R-loop can form transiently and can return back to the duplexed form.

Second, 0.9 state is only reached after 0.4, indicating that the R-loop is required to form G4. Third, the steady 0.9 state reveals that the G4/R-loop state is a stable conformation, which does not return to the other two states. The high FRET state remains after RNase H digest in FRET2 and FRET3, respectively, indicating that the G4 persist even after the R-loop is removed. To distinguish the G4 from other DNA, we ran a potassium polyacrylamide gel electrophoresis (PAGE) gel (10% PAGE with

**Fig. 5 PQS orientation regulates R-loop formation and is detected in single-molecule FRET. a** Experimental scheme of smFRET2 construct. The Cy3 and Cy5 are labeled across PQS region to detect transcription bubble formation (gray arrow) through PQS. **b** Trace examples demonstrate DNA only and the pattern of transcription events. **c** The transcription rate is calculated from the frequency of smFRET2 pattern in **b**. Data are presented as values ± SEM. $N$ value indicates the number of independent events. Data processing is described in "Methods," and statistics is reported in Supplementary Table 3.8. Source data are provided in Source Data file. *$P < 0.1$, ***$P < 0.001$, NS: nonsignificant (one-sided Kruskal–Wallis test). **d** FRET histograms with 10 min incubation interval shows an additional 0.1 FRET growing. Shown is a single representative histogram from $n = 5$ independent FRET measurements for each time point and condition. Histograms are fitted by Gaussian function and the data are provided in Supplementary Table 3.17. **e** RNase H removes 0.1 FRET peak, indicating an R-loop structure. **f** DNA substrate shifts to higher position, indicating an unexpected structure appearing during transcription. RNase H-mediated removal indicates that the structure is R-loop. Single representative image from $n = 3$ independent measurements. **g** The correlation between fraction of 0.1 FRET and R-loop peak. The fitted numbers of each spot are from Supplementary Fig. 4 and the data are provided in Supplementary Table 3.16. The correlation coefficients are 0.97, 0.92, and 0.96 for control, template, and non-template, respectively. Data are presented as mean ± SEM of $n = 3$ and 5 for R-looped band and 0.1 peak fraction, respectively. **h** RNAP titration for each construct indicates that R-loop formation is RNAP dependent. Shown is a single representative image of $n = 4$ independent experiments and is taken by Cy5 emission. Source data of full scan image and molecular marker is stained separately and provided as Source Data file. **i** Fraction of R-looped band fitted from **h**. Data are presented as mean ± SEM of $n = 4$ independent experiments and provided in Supplementary Table 3.9.

100 mM KCl). The up-shifted R-loop band, after the RNase H treatment, yields the linear DNA band and an additional band at the position of the pre-formed G4 DNA, confirming the G4 formation during transcription, which is stable even after R-loop removal (Fig. 7e).

To test the function of G4 folded state more directly, we conducted RNAP and RNase H experiment on pre-formed G4 in FRET2 and FRET3. Expectedly, both showed a single 0.9 FRET, displaying G4 formation. Upon RNAP reaction, the FRET2 shifted to 0.1 and shifted back to 0.9 after RNase H treatment, indicating that the 0.1 state prior to RNase H treatment was G4/R-loop state. FRET3 peak remained at high FRET in both conditions (Supplementary Fig. 5a), indicating that the pre-formed G4 in NT remains stably folded, unperturbed by transcription and RNase H digestion. Consistently, the EMSA analysis also shows that almost 100% DNA becomes R-loop state in pre-formed G4 template after 10 min of RNAP reaction and the G4 persisted after the RNase H digestion (Supplementary Fig. 5b). Both data strongly suggest that G4 (NT) folded state is poised to induce R-loop upon RNAP reaction, leading to G4/R-loop state and such G4 is highly stable, resistant to RNase H digest. The RNA production tested on pre-formed G4 of PQS-T and PQS-NT indicated that pre-formed G4 in template fully blocks transcription, while the G4 in NT leads to active transcription (Supplementary Figs. 5c and 6). Importantly, the pre-formed G4 of PQS-NT in both FRET2 and FRET3 constructs, which became 100% G4/R-loop conformation showed robust mRNA production, further supporting the role of G4/R-loop as the transcriptionally proficient structure responsible for enhanced mRNA production in PQS-NT (Supplementary Fig. 7).

To directly test the role of R-loop in G4 formation and transcription, GTP was substituted by inosine triphosphate (ITP), which inhibits R-loop formation since inosine forms less stable base-pair with cytosine (Fig. 7g). The EMSA gel showed a complete disappearance of the R-loop band (up-shifted) bands under ITP condition, indicating that the up-shifted band is indeed the R-loop structure. In single-molecule experiments (Supplementary Fig. 8b) conducted in ITP condition, the 0.1 peak in FRET2 and 0.4, 0.9 peaks in FRET3 all disappeared, reflecting that without R-loop, G4 structure cannot form. This is consistent with our observation that G4 folding depends on the R-loop formation.

**Stable G4/R-loop structure enhances RNA production.** So far, we have demonstrated that transcription by RNAP on PQS-NT induces R-loop formation, which in turn leads to G4 formation in NT strand. Accordingly, removal of R-loop by ITP incorporation led to disappearance of G4. Next, we asked how the R-loop and

G4 structures contribute to RNA production. To modulate the degree of G4 formation, we varied buffer conditions from G4 destabilizing to G4 stabilizing conditions in the order of no monovalent cation, LiCl, KCl, NMM (G4 stabilizing ligand) and pre-formed G4. First, we quantified the fraction of G4 by FRET3 histogram, which displayed the expected pattern of increasing G4 formation as a function of G4 stabilizing conditions (Fig. 8a). Next, we performed EMSA based transcription assay which allowed for measurement of R-loop and RNA production in varying buffer conditions (Supplementary Fig. 9). We plotted the fraction of R-loop and amount of RNA in the order of G4 stabilizing conditions, referring to the level of G4 folding state in transcription by FRET3 (Fig. 8b–d). The graph shows that (i) the R-loop and G4 formation are correlated for PQS-NT, (ii) RNA production of PQS-NT is correlated with both G4 and R-loop; (iii) RNA production of PQS-T is not impacted by G4 or R-loop, and (iv) all differences completely disappear with ITP treatment, which abolish R-loop formation (Fig. 8d).

In summary, 5′-UTR PQS influences transcription level through forming R-loop, which leads G4 folding that stabilizes R-loop (Fig. 8e). All the transcription processes can potentially induce R-loop as shown by non-PQS control (Fig. 8e). Both PQS-NT and PQS-T induce G4 formation on NT and T, respectively, yet with opposite consequences. The G4 on PQS-NT stabilizes R-loop by G4/R-loop structure and produces more RNA; by contrast, the G4 in PQS-T blocks transcription and reduce the overall RNA production. Such contrast in orientation-dependent G4 effect was accentuated when G4 was pre-formed. The pre-formed G4 in PQS-NT yielded the highest amount of RNA, whereas pre-formed G4 in PQS-T produced no RNA (Fig. 8c).

## Discussion

Here we employed ensemble transcription readout, EMSA, and single-molecule assays to demonstrate the impact of PQS in transcription and elucidate the underlying mechanism. All our results indicate that PQS-NT located downstream of TSS leads to upregulation of mRNA production (Fig. 1) mediated by elevated frequency of transcription (Fig. 4) and higher degree of R-loop formation (Fig. 5) stabilized by the formation of G4 on the NT strand (Fig. 7). We show that R-loop promotes transcription by a mechanism that involves successive R-loop formation (Fig. 6). The propensity for R-loop formation is greatly increased by PQS-NT due to the subsequent G4 formation in the NT strand, resulting in enhanced transcription (Fig. 8).

Based on our result, about 30% PQS-NT folds into a stable G4 structure during transcription, but why the effect is not correlated to PQS sequence appeared to be puzzling as the sequence controls G4 folding pattern. By contrast, the orientation effect is

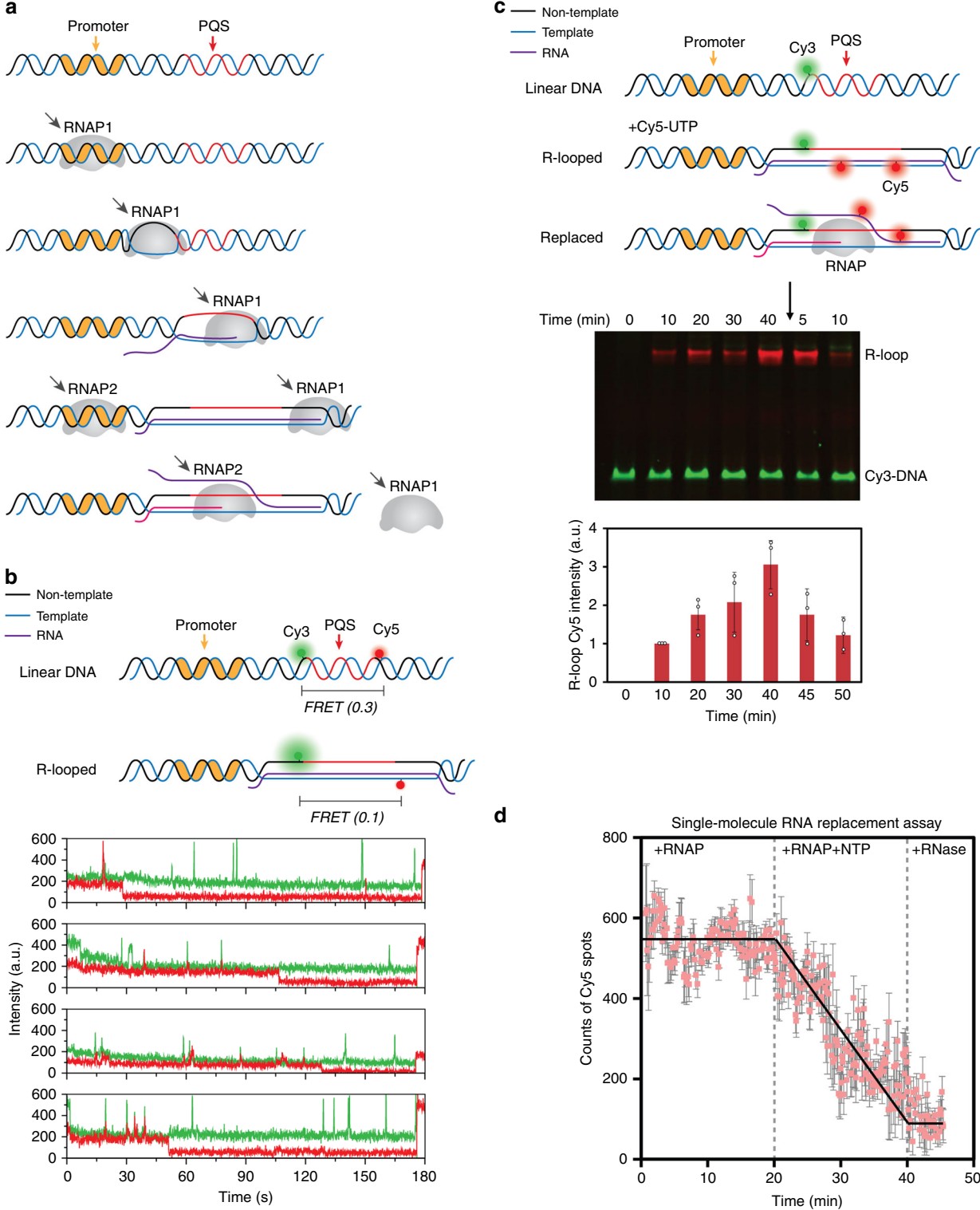

**Fig. 6 R-looped state is transcription active state. a** The proposed model to show how R-loop is synthesized and replaced during transcription. **b** Scheme (top) and trace examples (bottom) show that the transition from 0.4 state to 0.1 state is not caused by photobleaching, which is confirmed by switching to 641 nm (red) laser to excite Cy5 at last 50 frames of each movie. In addition, the PIFE signal of Cy3 indicates the transcription event still continues. **c** Scheme of single-molecule replacement assay (top). Cy3-labeled DNA construct is incubated with Cy5-UTP and other NTP mix. The color of Cy3 and Cy5 are used to visualize DNA (Cy3) and R-loop RNA (Cy5) in the EMSA gel (middle). Fresh non-Cy5-UTP is added after 40 min incubation and replaces the Cy5 R-loop, which causes the decrease of Cy5 signal (bottom). Shown in the middle is a single representative image of $n = 3$ independent experiments, presented by overlaying Cy3 and Cy5 emission. Source data of full scan image and molecular marker is stained separately and provided as Source Data file. Quantitative data are presented as mean ± SME and exact values are provides in Supplementary Table 3.10. **d** Cy5 counts from sm-replacement assay shows decrease of Cy5 spots. Each stages are negative control (RNAP only), test (fresh RNAP with non-Cy5 NTP), and final digestion by RNase H.

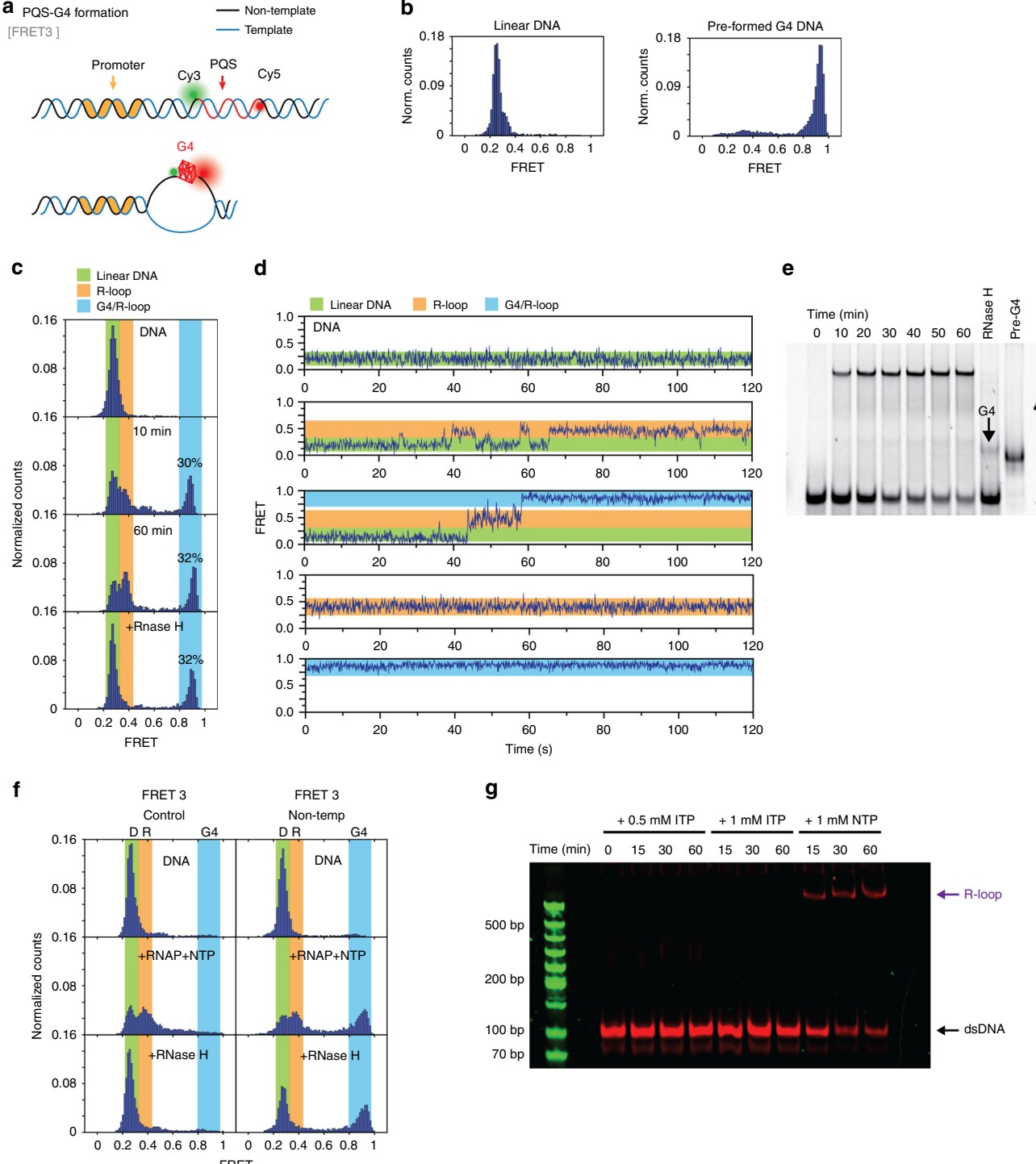

**Fig. 7 G4 forming during transcription is detected by single-molecule FRET. a** Scheme shows how FRET3 construct can probe the formation of G4. **b** The FRET histogram shows that both linear dsDNA and pre-formed G4 DNA have one sharp peak at 0.25 and 0.9, respectively. **c** FRET peak (0.25) transits to higher FRET population, 0.4 and 0.9 during transcription, indicating the formation of R-loop and G4/R-loop. **d** Trace examples of FRET3 shows three distinguishable FRET states. The transition occurs between 0.25 and 0.4, indicating the conversion of linear DNA to R-loop state. Transition from 0.4 to 0.9 indicates the formation of G4 structure requires R-loop forming first. **e** Similar to **c**, the G4 formation during transcription can be visualized through removal of hybrid DNA:RNA by RNase H treatment. Here, a 10% PAGE KCl gel (100 mM) is used to distinguish G4 DNA from linear DNA. The image is a representative gel from $n = 2$ independent experiments and taken under Cy5 emission. Source data of full scan image and molecular marker is stained separately and provided as Source Data file. **f** FRET3 control and non-template test shows the 0.4 is R-loop state and 0.9 is G4 structure. There are only two peaks in control, indicating no G4 structure. The 0.4 peak is R-loop, demonstrated by RNase H digestion. D is linear DNA; R is R-loop; G4 is G4/R-loop. **g** ITP substitution prohibits R-loop formation and causes disappearance of R-loop bands. The image is a representative gel from $n = 2$ independent experiments. Source data of full scan image is stained separately and provides as Source Data file.

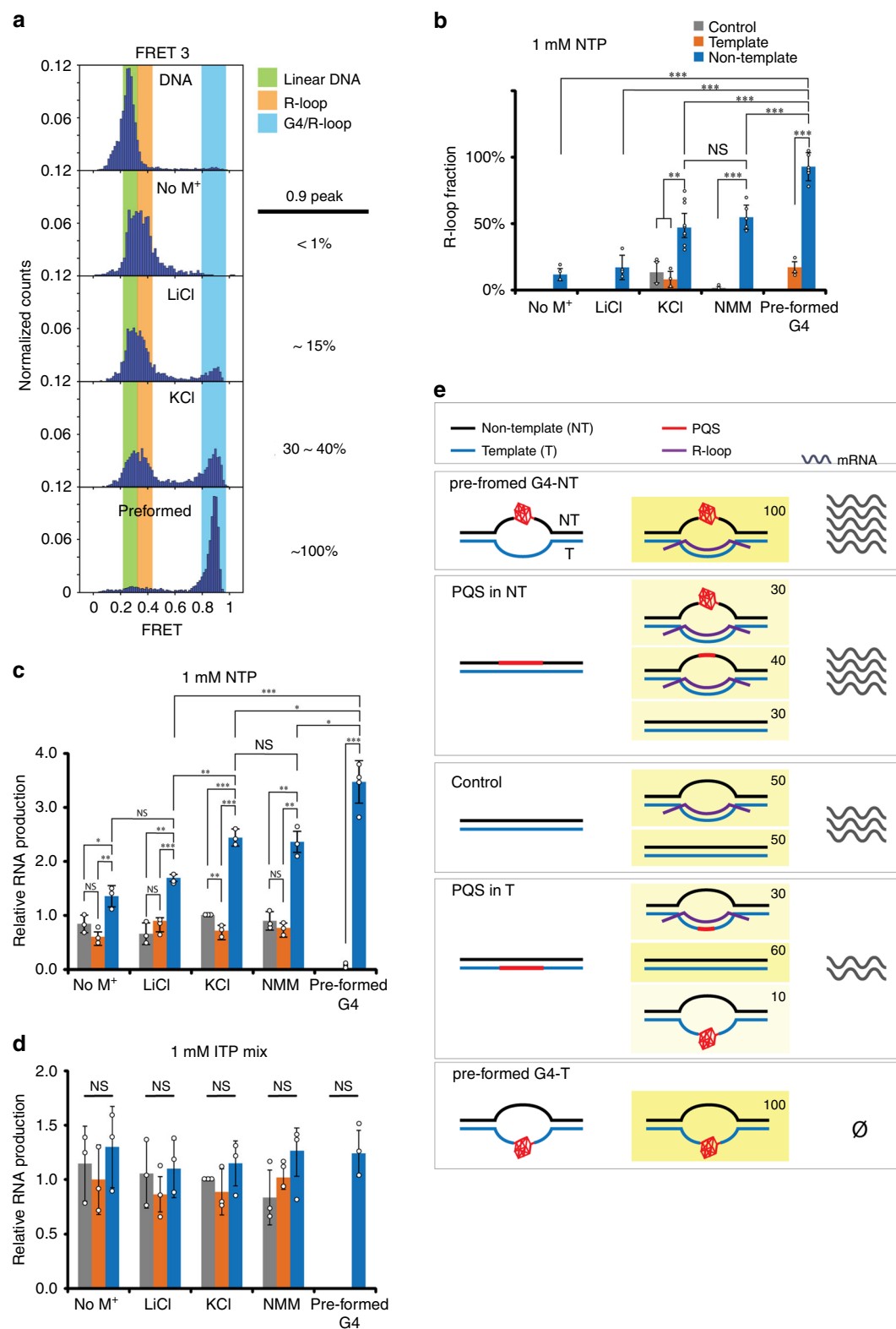

significantly more dominant in transcription as evidenced by all PQS-NT displaying higher mRNA level than the non-PQS control and all PQS-T (Fig. 1e), although some of the PQS are weak or unable to form G4 structure on its own[30,49]. This indicates that G4 is likely not the main driving force in regulating transcription, but PQS orientation matters more. Our results strongly suggest that R-loop is responsible for enhancing transcription. Indeed, R-loop forms first, followed by G4 even for the strongest G4 folding

sequence, cMyc used all throughout the single-molecule and EMSA analysis. Nevertheless, the transcription-induced R-loop formation may also depend on the position, length, and composition of G-rich sequence in the NT strand, which warrants future investigations.

By using the FRET3 construct, we demonstrated two versions of R-looped state including the R-loop without G4 (0.4) and the G4/R-loop (0.9) in which the R-loop gives rise to G4/R-loop

**Fig. 8 Correlation of G4, R-loop structure, and RNA production. a** FRET histograms of FRET3-NT with different G4-stabilizing conditions. The order is presented from weak to strong: non-monovalent, LiCl, KCl, pre-formed G4. The fraction of 0.9 peak is fitted by Gaussian distribution. **b–d** Plot of R-loop fraction and RNA production quantified from gel, Supplementary Fig. 9. The order is determined from the G4 fraction in **a**. **b** R-loop fraction of each condition in the presence of 1 mM NTP mix. **c** Relative RNA production in the presence of NTP, which is normalized to control (50 mM KCl). **d** Relative RNA production in the presence of 1 mM ITP mix. In the presence of NTP, PQS-NT shows a high correlation but PQS-T shows no difference among different G4 conditions. In the presence of ITP, the differences were eliminated due to the lack of R-loop and G4 structure. For **b–d**, data are presented as mean ± SEM of independent experiments ($n = 3$–11). *$P < 0.05$, **$P < 0.005$, ***$P < 0.0005$, NS: nonsignificant (two-sided unpaired $t$-test). Data and exact $P$-values are provides in Supplementary Tables 3.11–13. **e** Summary schematic of PQS effect on transcription. Pre-formed G4-NT leads to 100% G4/R-loop and enhances RNA production, but pre-formed G4-T causes transcription blockage. PQS-NT upregulates the transcription by additional G4/R-loop state, whereas PQS-T causes less R-loop formation due to the competition of G4 forming in template strand.

structure (Fig. 7c, f). Thus, the non-basepaired NT DNA strand that results from the R-loop can fold into G4 structure, which is indistinguishable in FRET2. When R-loop forms in FRET2, the dye-to-dye distance is out of FRET-sensitive range, likely due to the increased distance between the two dyes when the transcription bubble is opening; therefore, G4 formation in NT cannot be detected (Supplementary Fig. 8a). We also noticed that the G4/R-loop (0.9 FRET) forms primarily in the initial 10 min of transcription reaction, likely when the RNAP activity is maximal at the highest available NTP concentration. Once formed, G4 may stabilize the R-loop by effectively preventing the duplex DNA formation and/or facilitating the movement of RNAP on template. Interestingly, the RNase H digestion induces 0.4 state to return to duplexed DNA (0.25), yet the G4 at 0.9 state persists (Fig. 7c, bottom). The EMSA gel shows that the upper band (slower mobility) is removed after RNase H treatment, indicating that the upper band contains both R-loops, 0.4 and 0.9 states (Figs. 5f and 7e). Therefore, the higher level of transcription enabled by R-loop in PQS-NT is due to the additional G4/R-loop that cannot form in control or PQS-T (Fig. 5d).

Our results indicate that the R-loop induces continuous transcription activity despite the high thermal stability. One possibility is that the R-loop presents itself as a pre-open state for RNAP to access the T strand with a lower energetic barrier than the duplex DNA. In this way, the R-loop can serve as the structural conduit for increased RNA production (Figs. 1, 5d, and 8c). In agreement with previous study, we observed that PQS in T form G4 during transcription and inhibit transcription[50] (Supplementary Fig. 6). Collectively, transcription of PQS containing sequence can induce R-loop and G4. PQS-NT promotes transcription by G4/R-loop formation, whereas PQS-T diminishes transcription by negligible R-loop and G4 on template, which blocks RNAP.

Importantly, R-loop formation depends on the concentration of RNAP (Fig. 5h, i), in agreement with a previous study[27]. Indeed, both smFRET and EMSA assay show the R-loop formation is saturated within 30 min of active transcription (Supplementary Fig. 4d, e). Also, our replacement assay is measured under fresh NTP condition in which R-loop undergoes active transcription. After NTP is consumed, transcription rapidly slows down, likely making RNAP less capable of replacing R-loop and causing transcriptional decrease. Previous studies reported both up- or downregulation of transcription and translation led by different PQS orientations and such conflicting results may arise from differences in experimental design including position of PQS with respect to promoter, distance from the TSS, length, and composition of G-rich sequence[7,8,20,27,50]. Further, we tested the position-dependent effect by varying the distance between the promoter and the PQS region at 9, 16, and 41 bp. The transcription reaction was analyzed by EMSA gel and the quantification of R-loop and RNA product was plotted (Supplementary Fig. 10). Interestingly, our result shows that RNA production is reduced when the PQS is close to the promoter (9 bp), despite the

high R-loop fraction. This indicates the effect of R-loop depends on the position from the promoter.

Combining smPIFE and smFRET assay, we resolved dynamics of transcription coupled with the changes in DNA structure, providing a potential structure–function relationship between R-loop, G4, and transcription at the molecular level. Together, our results reveal the intrinsic effect of PQS in both orientations and the persistent, yet dynamic formation of R-loop and G4 structures, which lead to enhanced transcription. In light of the extensive content of PQS in genomic DNA, we propose that the R-loop and G4 will be a prevalent and important structure for transcriptional regulation in cells. In the cellular context, these structures may form transiently due to cellular proteins such as RNase H and G4 helicases that can disrupt the R-loop and G4, respectively and ssDNA-binding proteins that can compete with G4 formation. Such effect may be counterbalanced by factors that may stabilize G4 and R-loop. Another plausible scenario is that the PQS can function as a molecular switch that only activates in transcriptional burst, as these structures only form under actively transcribing condition.

## Methods

**DNA preparation for ensemble and single-molecule beacon assay.** DNA samples were PCR-amplified from a lab-modified plasmid, which is constructed by replacing the original promoter and 5′-UTR region of pZEMB8[51]. First of all, a linear DNA oligonucleotide purchased from Integrated DNA Technologies (IDT) was designed with a T7 promoter and an additional insertion site for PQS DNA (Supplementary Table 1) and then cloned upstream of a *GFP* gene sequence by restriction and ligation protocols. Next, different PQS DNA samples (Supplementary Table 2) were inserted into the 5′-UTR site through the same protocol. Finally, biotinylated forward primer and T7 terminator reverse primer (Supplementary Table 1) were used to amplify linear DNA, which contained biotin, T7 promoter, 5′-UTR PQS, *GFP* gene, and T7 terminator. The biotin was used to immobilize DNA on a PEG slide.

**DNA preparation for single-molecule PIFE and FRET assay.** The DNA oligonucleotides (Supplementary Table 1) were purchased from IDT with an internal amine modification, which was used for Cy3 or Cy5 labeling. The labeled NT and T DNA were annealed with biotinylated 18-mer primer (Supplementary Table 1) in 10 mM Tris buffer (pH 8.3) at the ratio 1 : 1.2 : 1.5, respectively. The mixtures were heated at 95 °C for 5 min and slowly cooled to room temperature (1 °C per min).

**DNA labeling for single-molecule samples.** The amine-modified DNA oligonucleotides were dissolved in $H_2O$ first to obtain 100 μM stock. DNA (25 μL, final concentration 50 μM) was mixed with 0.1 mg Cy3- or Cy5- NHS-ester (GE Healthcare), 5 μL 1 M sodium bicarbonate buffer (freshly prepared), and 25 μL $H_2O$. The reaction was kept in the dark and rotated overnight at room temperature. The excess dye was removed from DNA sample by running ethanol precipitation twice. For ethanol precipitation, the 50 μL reaction was mixed with 125 μL 100% cold ethanol and 3 μL 5 M sodium chloride solution, and then cooled at −80 °C for 1 h. The frozen sample was centrifuged at 4 °C with $21,120 \times g$ (or at least 15,000 r.p.m.) for 30 min. The pellet was washed twice with 70% cold ethanol. After twice ethanol precipitation, the DNA sample was suspended in 25 μL 10 mM Tris buffer.

**In vitro T7 transcription assay.** Ensemble in vitro transcription assay was performed by electrophoresis and TECAN Spark plate reader at 37 °C. Each sample was prepared with 1 nM DNA template in transcription buffer (40 mM Tris-HCl

pH 8.3, 50 mM KCl, 6 mM $Mg_2Cl$, 2 mM spermidine, 1 mM dithiothreitol) and mixed with RNase inhibitor murine (0.4 unit/µL), inorganic pyrophosphatase (0.02 unit/µL), and T7 RNA polymerase (1.25 unit/µL). The reaction was initiated by adding NTP mix for a final concentration of 1 mM. For electrophoresis, reactions (20 µL) were quenched by 0.5 µL of 0.5 M EDTA at each time point and post-mixed with 100 nM molecular beacon probe (Supplementary Table 1). The samples were run on a 3% agarose gel. The image was taken by gel imager (Amersham imager 600) with 520 nm LED light. For the plate reader, each reaction (100 µL) was pre-mixed with 100 nM probe and loaded on 96-well transparent plate (Thermo Fisher Scientific). The data points were collected per minute at $\lambda_{ex.}$ 545 nm (slit size 10 nm) and $\lambda_{em.}$ 570 nm (slit size 10 nm). The RNA production in gel was quantified by ImageJ. The linear portions (first 20 min) of Cy3 intensity curves were used to fit the rate of Cy3 increase and the rate was normalized to non-PQS control.

**Electrophoretic mobility shift assay**. All the gels with single-molecule DNA samples were performed by using 10% PAGE gel. Each sample (20 µL) was prepared with 10 nM labeled DNA in transcription buffer and mixed with RNase inhibitor murine (0.4 unit/µL), inorganic pyrophosphatase (0.02 unit/µL), and T7 RNA polymerase (1.25 unit/µL). The reaction was initiated by adding NTP mix or Cy5-UTP and incubated at 25 °C, and quenched by 0.5 µL of 0.5 M EDTA at each time point. For RNase-treated sample, the reaction was first stopped by adding 1 µM 22-mer T7 promoter DNA (Supplementary Table 1). Next, the samples were added with 0.5 µL RNase H (final concentration 0.125 U/µL), incubating at 37 °C for 15 min, and terminated by EDTA. For GTP substitution, 1 mM ATP, CTP, UTP, 1 mM ITP, and 4 mM GMP (guanosine monophosphate) were added to initiate reaction[26]. For G4 stabilization test, 50 mM KCl was removed or replaced by 50 mM LiCl and 1 µM NMM was added selectively. The samples were mixed with 4 µL 50% glycerol and 0.1% SDS before loading on the gel. The gel was run at a constant 10 mA and stained by Sybr-Green II RNA gel stain. The image was taken by gel imager (Amersham imager 600) with 520 nm and 630 nm LED light. For KCl-PAGE gel, both the gel and running buffer were added additional 100 mM KCl. The gel was run at a constant 100 V in the cold room. All the gels were quantified by ImageJ to obtain the fraction of DNA and the amount of RNA.

**Gel quantification**. All the gels were quantified by ImageJ and processed by a two-step normalization. For example, R-loop fraction was calculated as the percentage of R-loop band at the same lane, indicating it was normalized by total DNA signal. For RNA replacement assay, RNA (red, Cy5) signal was quantified and normalized to the DNA (green, Cy3) signal at the same lane. As the DNA samples were prepared from a master mix, the addition signal at each lane represented RNA signal after normalization. For RNA production, Cy5-only DNA was used to visualize DNA and RNA was stained by Sybr-Green II (Invitrogen), which emits fluorescence in Cy3 (green) channel. The RNA (green) signal was normalized by DNA (red) in order to compare RNA production with the same amount of DNA. To achieve this, the labeling efficiency of Cy5 should be similar and high enough to represent DNA concentration. Here, the labeling efficiencies are 96%, 95%, and 95% for control, T, and NT, respectively. Furthermore, to compare the relative RNA production among different constructs and buffer conditions, the normalized RNA signal was normalized against the control (50 mM KCl) in the same gel and plotted in bar graphs.

**Single-molecule assay**. All the single-molecule assays were performed by using a home-built prism-type total internal reflection fluorescence microscope at room temperature (23.0 ± 1.0 °C)[34,52]. DNA stock (10 nM) was diluted to 25 pM and immobilized on a PEG slide pretreated with neutravidin (0.05 mg/mL). The imaging buffer used for single-molecule measurement was prepared freshly by mixing transcription buffer with an oxygen scavenging system (1 mg/mL glucose oxidase, 0.8% v/v glucose, ~10 mM Trolox, and 0.03 mg/mL catalase). Solid-state 532 nm and 641 nm lasers were used for single-molecule measurement. Single-molecule traces were recorded with a 100 ms time resolution by smCamera software and analyzed with Interactive Data Language. The trace outputs were processed with custom MATLAB script to generate trajectories and FRET histograms. Each FRET histogram was generated by collecting FRET values from at least 6000 molecules taken over 15~20 movies with the removal of donor (Cy3) only containing signal and analyzed with Gaussian distribution function.

**Single-molecule beacon assay**. The experimental design was adapted from a previous publication[53]. DNA samples prepared for ensemble assays were diluted and immobilized on PEG slide. The reaction buffer was freshly prepared imaging buffer with RNAP (1.25 units/µL), NTP mix (100 µM or 1 mM), and 100 nM beacon probe. Buffer was injected into the measurement chamber through a home-made flow system. A 532 nm laser was used to excite Cy3 dyes and each measurement was collected as a 6 min movie. Single Cy3 burst steps were counted from over 100 traces and the event rate was calculated from the dwell time between each burst.

**Single-molecule PIFE and FRET assay for binding and initiation**. DNA samples for both binding and initiation assays were prepared from the same substrates, FRET1-Cy3 and FRET1-Cy5 (Supplementary Table 1). Binding substrate was labeled with Cy3 on the NT strand; initiation substrate was labeled with Cy3 and Cy5 on each strand. A 532 nm laser was used to excite the DNA sample. RNAP concentration was calibrated by a standard bovine serum albumin sample. For the binding assay, imaging buffer was mixed with titrated RNAP of 10, 50, and 100 nM as the final concentration. The trace data were analyzed to calculate transcription rate (frequency of PIFE signals) and average number of events. For the initiation assay, imaging buffer was mixed with 100 nM RNAP and titrated NTP mix of 10 µM, 100 µM, and 1 mM as the final concentrations. The trace data were analyzed to calculate the transcription rate (frequency of FRET bursts). Fraction of initiation was calculated by dividing the number of initiation events with the number of total events.

**Dual-PIFE assay for elongation**. DNA samples for dual-PIFE assay were annealed with FRET1-Cy3 and FRET2-Cy5 (Supplementary Table 1). The imaging buffer was mixed with 100 nM RNAP and 100 µM NTP mix. Both 532 nm (green) and 641 nm (red) laser were used to excite Cy3 and Cy5 simultaneously. Fraction of elongation was calculated by dividing the number of dual-PIFE events by the number of total PIFE events.

**Single-molecule FRET assay for elongation**. DNA samples were annealed with FRET2-Cy3 and FRET2-Cy5 (Supplementary Table 1). The imaging buffer was mixed with 100 nM RNAP and 100 µM NTP mix. Transcription rate was calculated from the frequency of FRET events. For the R-loop FRET histogram, the imaging buffer was mixed with 1 mM RNAP and 1 mM NTP mix. Each FRET histogram was generated from 20 movies with over 6000 molecules total. For RNase treatment, the imaging buffer was mixed with RNase H (final concentration 0.05 U/µL).

**R-loop replacement assay**. DNA samples were annealed with FRET2-Cy3 and FRET2-Cy5 (Supplementary Table 1), but only the Cy3 strand was labeled. This assay was tested by EMSA and single-molecule measurements. For EMSA assay, the transcription reactions were incubated with 10 µM Cy5-UTP mixtures with 1 mM ATP, CTP, GTP, and then an additional 2 mM NTP mixture (non-Cy5-UTP) was added at 40 min time point. Next, a 10% PAGE gel was used to differentiate linear DNA and R-looped DNA, where DNA and RNA were visualized in green (Cy3) and red (Cy5), respectively. The Cy5 bands were quantified as Cy5-incorporated RNA. For single-molecule measurement, the transcription reaction were pre-incubated for 30 min and then immobilized on PEG slide. Both 532 nm (green) and 641 nm (red) laser were used to excite Cy3 (DNA) and Cy5 (RNA), and the molecules were selected with both Cy3 and Cy5 colocalized. RNAP (1 mM) without NTP was injected into observation chamber as control experiment. Replacement was initiated by flowing 1 mM RNAP and 1 mM non-Cy5 NTP mixture. One hundred and fifty short movies were randomly collected from whole slide area over 20 min. Each movie was taken only 2 s to avoid photobleaching. The number of Cy5 spots was quantified from each short movie and plotted as function of time.

**Single-molecule FRET assay for G4 formation**. The DNA sample was annealed with FRET3-top and FRET3-bottom (Supplementary Table 1). The FRET3-top strand was labeled with both Cy3 and Cy5 dyes. Pre-formed G4 DNA was annealed in 40% PEG-200 and 100 mM KCl with the same heating and cooling protocol. The imaging buffer was mixed with 1 mM RNAP and 1 mM NTP mix or ITP mix. Each long movie was collected for 2 min. Each FRET histogram was generated from 20 movies with over 6000 molecules total. For RNase treatment, the imaging buffer was mixed with RNase H (final concentration 0.05 U/µL).

**Statistics analysis for smFRET**. Data shown in Figs. 1h, 3c, g, and 5c were obtained from individual and independent events. All numbers were convoluted to a continuous lifetime curve, and the mean lifetime (value ± SEM) was calculated by fitting to first-order exponential decay function. As the convoluted curve is not normal distribution, the statistic test was performed by Kruskal–Wallis test, a nonparametric statistics method. Mean lifetime calculated as rate (or frequency) and number of events were reported in Supplementary Tables 3.2, 3.3, 3.5, and 3.8. Data shown in Fig. 3d were counted from individual trace (independent molecules). The average number and statistics test was performed with two-sided unpaired t-test, reporting in Supplementary Table 3.4. All the raw data were provided as a Source data file.

**Reporting summary**. Further information on experimental design is available in the Nature Research Reporting Summary linked to this paper.

## Data availability
The datasets generated during and/or analysed during the current study are available from the corresponding author on reasonable request. Source data are provided with this paper.

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

## Acknowledgements

This work was supported by the National Institute of Health General Medicine (1R01GM115631-01A1) to C.L. and S.M., National Science Foundation Physics Frontiers Center Program (0822613) through the Center for the Physics of Living Cells to S.M. We

thank Frank Hua for careful reading of the manuscript and the members of the Sua Myong and Taekjip Ha laboratory for helpful comments.

## Author contributions

All single-molecule experiments were performed by C.L. and C.M. with the assistance provided by K.M., W.Z., and A.W. The manuscript was written by S.M. and C.L. All authors have given approval to the final version of the manuscript.

## Competing interests

The authors declare no competing interests.
