## [Peer Review File · Nature Communications]

Reviewers' comments:

Reviewer #1 (Remarks to the Author):

The Myong laboratory studied the transcriptional output of a model gene harboring potential G-quadruplex forming sequences (PQSs) of varying length on the template or non-template strand of the 5'-untranslated region during T7 RNA polymerase-mediated transcription. They found the presence of a PQS on the non-template strand led to more efficient transcription, regardless of length, than the case with the variable PQSs on the template strand. The authors conclude an R loop structure that forms when the PQS is on the non-template strand is responsible for the effect observed.

One set of control experiments should be conducted to support the R loop. Transcription efficiencies should be conducted with inosine triphosphate (R loop destabilizer) or 7-deazaguanosine triphosphate (G4 destabilizer for the transcript) to support the R loop claim and the role of a G4 on the mRNA. A recipe for these experiments was previously reported by the Hanawalt lab (PNAS, 2010, 107, 12816). Indeed, the Hanawalt publication should be referenced in any case since it leads to similar conclusions.

Various aspects of the introduction and conclusions are confusing, including some ambiguity about whether this manuscript and the papers referenced are talking about G4s in untranscribed promoters or in 5'UTRs. For example, G4s in promoters have been studied in various locations to give increases in gene expression from both strands (Fleming et al, NAR, 2019). Also, the work of Fleming and Burrows who studied strand dependence (up and down, correlating with the work here) was not referenced at all, and they have several papers on this starting with PNAS in 2017, and more specifically for up/down strand dependence, in ACS Chem. Biol. 2017 in the case of oxidative damage to the PQS.

More labeling is needed for readers to decipher the figures. What are the red strands? What are the orange-filled duplex regions? What do the gray arrows indicate? The figure captions need to spell out these points that are only sometimes labeled.

I think there is interesting data here, but it needs to have the English polished a little bit and then to make stronger conclusions regarding the impact of a folded G4. For example, the changes seen for template vs. non-template effects appear to be small, if I am correctly reading Figs. 1e, 4d and 5c.

Reviewer #2 (Remarks to the Author):

I have submitted my report in the attached Review.pdf file.

Reviewer #3 (Remarks to the Author):

The work by Lee et al. assesses the potential role of G-quadruplex (G4) in promoting transcription. Using a combination of bulk and single-molecule experiments the authors show that the presence of a G4-forming sequence downstream a TSS and on the non-template strand stimulates transcription. The main conclusion of the manuscript is that while the presence of the G4-forming sequence does not affect T7 RNA binding and transcription initiation, it affects the elongation step. The authors propose that G4 formation in the non-template strand favour R-loop formation that in turn stimulates transcription elongation. Because the exact mechanism by which G4 can promote or repress transcription are not well characterised, this manuscript is of potential interest for the broad readership of Nature Communication. Nevertheless some of the conclusions of the manuscript are not fully supported by the data and additional control experiments are needed to support the role of G4 in R-loop stabilisation. Here are some comments/suggestions:

Main points:

(1) The data reported in the current manuscript support that a guanine-rich sequence in a non-template DNA strand promotes transcription, which is in stark contrast with previous reports (Belotserkovskii et al. PNAS 2010, NAR 2012 and NAR 2017). The experimental setups in these works differ only by the position of the guanine-rich sequence compared to the TSS. Hence the position of the guanine-rich sequence seems crucial in determining its inhibitory or stimulatory effect. Because the novelty of the current manuscript lies in the stimulatory effect of the guanine-rich sequence it seems essential to study the positional effect of the guanine-rich sequence.

(2) The main conclusion of the article is that G4 folding within the non-template strand stabilises the R-loop. Nevertheless the authors did not demonstrate accurately this point and the data reported in Fig 1e suggest otherwise (e.g. the AAA-NT sequence which is not prone to G4 folding according to the NMM-based fluorescence assay stimulates transcription). To support the conclusion of the paper, the authors need to demonstrate that the structure on the non-template strand rather than the guanine-richness of the synthesised RNA stabilises the R-loop. The authors should perform the different transcription assays in the presence or the absence of KCl, using a control sequence of similar G-richness (e.g. point mutation in the c-myc sequence) or in the presence of a small molecule known to stabilise G4s to validate this point. These experiments are crucial since the inhibitory effect of R-loops reported by Belotserkovskii et al. is independent from G4 formation.

(3) The author concludes that the presence of the G4-forming sequence does not affect transcription initiation. Nevertheless the data reported in Supplementary Fig 2e shows that the presence of the G4-forming sequence decreases by $\sim 5\%$ the number of abortive events. Knowing that the G4-forming sequence in the non-template strand increases by $\sim 8\%$ the number of successful elongation events (Fig 4d), why is the effect on abortive events ruled out to explain the stimulatory effect of G4-forming sequences?

(4) It is surprising that the stimulatory effect of the R-loop is observed only for high concentrations of the RNA polymerase. One explanation could be that a high concentration of RNA polymerase is needed to displace the R-loop after the first round of transcription. The author could assess the impact of RNA polymerase concentration on the rate of R-loop displacement using their single-molecule RNA replacement assay reported in Fig 6.

Minor points:

(5) In order to appreciate the relevance of the differences between the G4-forming sequence in the template, non-template and control constructs (Fig 3d and h, Fig 4d, Fig 5c and corresponding supporting figures) the authors should use appropriate statistical tests and report their values.

(6) On figure Fig 1d the initial slope of the control sequence (black curve) seems to be bigger than the slopes for the non-template constructs which is not the case on the quantification reported in Fig 1e.

Review

Lee et al. have performed in vitro single molecule and bulk measurements on constructs that contain a G-quadruplex (G4) forming sequence in either the template or non-template strand of a double stranded DNA (dsDNA) that also contains an upstream promoter site for T7 RNA polymerase. They have studied the impact of having such potentially G4 forming sequences in the template (PQS-T) or non-template (PQS-NT) strand on transcription and R-loop formation. They propose that PQS-NT increases transcription production rate and yield. They attribute this increase to R-loop formation, which induces G4 in the non-template strand. This G4 is proposed to enable successive rounds of R-loop formation and increase transcription. The authors have performed single molecule FRET, protein-induced fluorescence enhancement (PIFE) and gel electrophoresis measurements to support their conclusions on this important topic. Despite a number of bulk in vitro and in vivo studies on the topic, there has not been a single molecule work that investigates this question (to the best of my knowledge), which is a novel aspect of the study. Prof. Myong and her lab have extensive experience on G4 structures and both the single molecule and bulk methods used in this study.

As the authors also cited, the impact of G4 forming sequences on R-loop formation has been demonstrated as early as 2004 using electron microscopy (Duquette....Maizels, Genes and Dev, 2004). Since then there has been a number of studies on this topic, including some whole genome studies that demonstrated high degrees of correlation between R-loop forming and G4-forming regions. In another recent study (also cited by authors) extensive in vivo studies were performed in different cells to investigate connection with G4 stabilization by ligands and R-loop formation (De Magis et al. PNAS 2019). In particular, the impact of non-template G4 structures on R-loop formation and the resulting DNA damage has been well-established. There are also studies that connect G4 structures within the template strand to transcription inhibition. Given this history, the manuscript by Lee et al. should be evaluated based on its contributions to our mechanistic understanding, which has been identified as the primary goal by the authors as well. My main criticism about the manuscript is that I do not find the proposed mechanism and link between G4 and transcription regulation convincing. I found the connection between G4 structures and R-loop formation to be better justified, but in my opinion, these are the already better known aspects of the field. My criticisms are as follows:

Major Criticisms:

1- The authors claim that their studies demonstrate diminished transcription when PQS is in the template strand. However, I do not think the assays that probe transcription show a significant difference between PQS-T and the Control cases in terms of the number of elongation events (Fig 4d), transcription rate (Fig 5c) or rate of Cy3 burst (Fig 1h), which is proposed to depend on the amount of transcribed RNA. Based on visual estimate of the error bars (I did not see them explicitly given), the following numbers are reported:

Fig. 1h: Rate for Control 1.3 ± 0.25 ; Rate for Template 1.2 ± 0.2

Figure 4d: % successful elongation for Control 50 ± 5 ; % for Template 45 ± 2

Figure 5c: Rate for elongation events for Control 3.4 ± 0.4 ; for Template 3.2 ± 0.2

None of these numbers are particularly conclusive in demonstrating a diminished transcription. The differences are more prominent when levels of R-loop formation are compared (0.1 FRET peak

population), but this does not mean diminished transcription. So, the claim for diminished transcription should be justified. Also, more prominent reductions in transcription have been reported for other systems when G4 is in the template strand. For example, a 3x enhancement in c-Myc expression was demonstrated when G4 forming sequence was eliminated by a single nucleotide mutation (Siddiqui-Jain..Hurley, PNAS 2002). Can the authors comment on why the impact is much smaller in their assays?

2- In essentially all the data in Figures 1,4, and 5, the rates for Control and PQS-T are very similar (within the error bars), while the rates for PQS-NT constructs are slightly higher than those for PQS-T. In Figures 4 and 5, the rates of PQS-NT and Control are also very similar considering the error bars associated with the measurements. In general, the difference between the quoted rates is marginal. On the other hand, the populations of the 0.1 FRET peak in Figure 5d are very different for these cases. In particular, the population of this peak for the non-template construct is 2-3 times higher than that for the template construct throughout the measurement. This peak is interpreted to represent R-loop formation, which is proposed to be the underlying mechanism for enhanced transcription in PQS-NT constructs compared to PQS-T constructs. The authors have not clarified the reasons for this quantitative discrepancy. If stable R-loop formation gives rise to higher levels of transcription, why isn't there a quantitative agreement between the two observations? One might even take the opposite perspective and propose that despite significantly higher levels of R-loop formation, the rates of transcription differ only marginally.

3- I found the evidence provided for successive R-loop formation to be weak and thought the single molecule and gel measurements were not coherent. For example, the authors suggest the spikes observed in Figure 6b to be evidence for successive rounds of R-loop formation. These jumps are ~10s apart and they take place at a constant 0.1 FRET base state. This would require not to have any time gap between one R-loop and the next, which is not clear how it could be possible. Also, in the EMSA it takes many minutes for Cy5-labeled UTP to be displaced by an unlabeled UTP, but in the single molecule traces the time difference is only ~10 s. It is not unexpected that given enough time, the RNA strand within an R-loop will be displaced by another RNA strand in a second round of transcription, which would result in replacement of the labeled UTP with the unlabeled one. But this is different from observing successive R-loop formation events within tens of seconds apart (5 events within ~120 seconds in Figure 6b). Why do the authors rule out simpler explanations such as successive binding of RNAP within the vicinity of Cy3? As shown in the traces in Figure 6b, such jumps take place even at the higher FRET state, before the transition to the 0.1 FRET state. Along these lines, I do not think Figure 6d also provides much support for the successive R-loop formation since it is expected that as RNA is synthesized the number of spots in the Cy5 channel will increase (more labeled UTP is incorporated) and when the chamber is treated with RNaseH these spots will be eliminated.

4- The molecular beacon in Figure 1 will be complementary not only to RNA but also to the non-template strand. So, the rise in Cy3 intensity (Figure 1c) could be due to not only binding of the beacon to RNA but also to the non-template strand. Later in the paper, the authors show that R-loop formation (between RNA and the template strand) is the dominant mechanism during transcription for these GC-rich sequences, particularly when G4 is in the non-template strand (PQS-NT). However, the analysis is performed assuming R-loops do not form, and transcribed RNA is available for binding to the molecular beacon. The authors should provide a justification for neglecting R-loops and their potential impacts on the binding characteristics of the beacon. This is particularly important since R-loops are not equally likely to form or remain stable for PQS-T and PQS-NT samples, and therefore, will impact the two constructs at different levels. The same criticism is valid for the gel analysis as well. If R-loop formation

impacts the availability of the RNA to hybridize with the beacon, then the beacon intensity does not necessarily correlate with the amount of transcribed RNA. These effects are complicated to decouple since it is not clear which segment of RNA takes part in R-loop formation, i.e. it is not known whether the segment that is complementary to the beacon takes part in R-loop formation.

Minor Criticisms:

1- The authors seem to suggest R-loop formation gives rise to G4 formation in the non-template strand, which in turn stabilizes the R-loop. Whether R-loop or G4 comes first is not clear, and I am not sure if there is any evidence in the manuscript supporting the proposal that R-loop formation comes before G4. Based on crystal structure, the RNA transcript and the template strand of DNA are known to come out of different channels of the RNA polymerase. This RNA transcript likely folds back on to the template strand to form the R-loop. However, during transcription and while the transcription bubble is open, the non-template strand should be relatively free to form G4, which makes the possibility of G4 forming before the R-loop to be potentially possible.

2- In Lines 46-48, the authors wrote: "Furthermore, there is no direct evidence to attribute the observed transcriptional effect of PQS to G4 formation rather than the guanine rich sequence itself." I don't think this statement is accurate since there are studies that directly probed the impact of G4 by mutating the sequence (including single nucleotide mutations) and demonstrated significant change in both mRNA and protein levels. The study from Hurley's lab, which was mentioned before, is an example for this (Siddiqui-Jain..Hurley, PNAS 2002).

3- Lines 97-98: Why does Cy3 show up as spikes? Is Cy3 photobleaching or dissociating from its complement? How do the authors distinguish Cy3 molecules that non-specifically bind to the surface from those that bind to the complement? 100 nM is much higher than typically used for donor labeled molecules in smFRET studies, but having a quencher has enabled performing the studies without excessive background. Nevertheless, this high concentration would result in significant transient non-specific binding to the surface, even a high-quality PEG surface. Even though the quencher suppresses Cy3 signal while the beacon is in the solution and flexible, it may not do so when it the beacon is transiently bound to the surface. In this immobile state, the end-to-end separation could be significant resulting in imperfect quenching. Have the authors performed a similar assay in an empty channel to estimate the level of such non-specific transient binding events?

4- In Figure 4c: What criteria did the authors use while counting the PIFE events? Even in these very nice traces, there are some events that may or may not be counted as PIFE events. For example, there two events around time=50s in the top trace of Figure 4c which are clear spikes in the in the Cy3 signal but are not as prominent as those shown with green arrows. There are similar events in other traces as well, which would require many judgment calls unless well-defined selection criteria are established.

5- Lines 150-153: the authors wrote: "Approximately 50% successful elongation was displayed in non-GQ control whereas 45% and 58% was observed for PQS-T and PQS-NT, respectively (Fig. 4d), strongly suggesting the higher rate of successful elongation in PQS-NT than the non-PQS and the PQS-T." Given the error bars (~5% in the control), the proximity of the cited averages, and the complex nature of the data analysis regarding selection criteria, I think "strongly suggesting" in this statement is not justified.

6- Lines 162-163, Figure 5d, and Fig 6b: In Figure 6b, the authors have shown the R-loop construct as linear while the dsDNA itself is in the form of a double helix. I understand that this is just a schematic, but it could be misleading since the R-loop will also have the double helix structure, probably close to the A-form. Since the A-form has shorter separation between the base pairs (although with a little greater diameter), one would expect a DNA-RNA hybrid to have a shorter end-to-end distance compared to the standard B-form dsDNA. However, the authors have proposed the R-loop to have FRET 0.1 while the dsDNA has FRET 0.3 for the same dye-positions. Can the authors elaborate on why the R-loop seems to have greater end-to-end distance compared to dsDNA? Does RNAP binding to an unwound dsDNA contribute to this lower FRET? Have the authors measured the FRET level that corresponds to RNAP binding to a transcription bubble that has Cy3-Cy5 at similar bp separation to their construct?

7- Lines 168-169: If the dsDNA is 90-bp, why does the R-loop run at 500-bp in the gel (Figure 5f)? Is this expected or reported before? Is RNAP expected to remain bound to the promoter on the dsDNA during this gel? It looks like for the RNaseH treatment, the RNAP is removed from the construct by adding high concentration of the T7 promoter before adding RNaseH. But it is not clear if RNAP is removed for the measurements before RNaseH. This question stems from the ambiguity about whether the shifted band at 500 bp could be due to binding of RNAP to the promoter on the DNA and not necessarily only due to the R-loop. A clarification of this would be helpful.

8- Line 191-192: the transition from 0.3 to 0.1 FRET is very fast as far as I can tell from the traces, possibly as fast as the time resolution of the measurements. Do the authors expect the proposed dsDNA to R-loop transition be this fast? The dual-PIFE measurements suggest that there are several second time difference between RNAP interacting with Cy3 and Cy5, which should set the lower time scale for the transition from duplex to R-loop, e.g. bubble creation, RNA transcription, and invasion of the DNA by RNA should take place before duplex to R-loop transition takes place. The other possibility is that the R-loop forms after the RNA is already transcribed and duplex reanneals. However, in that case it is interesting that none of the other dynamics mentioned above shows up in the traces and only a stable 0.3 FRET state is observed before R-loop formation is observed.

9- Line 223: Is the population of the 0.9 FRET peak only 20% of the total population at 60 min in Figure 7c? It appears to be much higher. Especially considering the population of the 0.9 FRET peak in Figure 5e is also quoted to be 20% of the population. The peak in Figure 7c appears to be 2-3x more populated than that in Figure 5e. On a related point, if an R-loop can form with or without the G4 in the non-template strand, why isn't there any R-loop with 0.9 FRET in Figure 5d? I understand that the Cy3 is moved from the template to the non-template strand however, this should not create a distance change large enough to reduce FRET 0.9 to 0.1. At least the R-loop peak remains at 0.1 FRET in both cases even though Cy3 is on the template strand in one case but on the non-template strand in the other. The 0.9 FRET peak appears only after RNaseH treatment (Figure 5e), which removes the RNA strand but does not influence the fluorophore positions. So if G4 can form (with or without the R-loop) in the FRET2 construct, it should be at FRET 0.9. Figure 7f mid-panel shows that the 0.9 FRET population persists even after R-loop formation, which suggests it should have appeared in Figure 5d as well. Given these, I am not convinced that the 0.1 FRET peak in Figure 5d represents R-loop with and without G4 formation.

10- Line 246: The authors wrote: "...the FRET2 entirely shifted to 0.1...." when describing middle panel in Figure 7f. However, a peak at 0.9 FRET is clearly visible in the histogram, so the shift does not seem to be complete.

Lines 247-249: What is meant by the statement “The results....RNaseH digestion.” is not clear to me.

Line 385: Is Trolox at ~10 mM concentration? Many labs quote ~ 2mM as saturating concentration in physiological pH buffers.

Figure 6e: In such analysis, typically the control for photobleaching is also demonstrated.

Figure 7f: It is surprising that no 0.4 FRET peak is observed in the Pre-formed G4 case for FRET3 construct after +RNAP+NTP are added. Given that state dominates in Figure 7c (when G4 is not preformed), I would have expected at least some population at that peak. Since effectively nothing has changed in the histograms for FRET3 construct after +RNAP+NTP or RNaseH treatment, do the authors know whether transcription took place in these FRET assays? As the R-loop readily forms and is very stable in all other assays, it is surprising that it does not form at all in this assay.

Typos: Line 55 (incomplete sentence), Line 58 (has/have), Line 62 (each stage), Line 123 (has/have), Line 143 (which puts), Line 168 (at 10-minute intervals), Line 610 (transcribed RNA), Line 611 (up traces?), Line 612 (as shown in f), Line613 (as shown in f).

**POINT BY POINT RESPONSE TO REVIEWERS' COMMENTS**
(Our responses are headed by left arrows, in navy blue)

We are extremely grateful for all of the reviewers' insightful criticisms and comments. We spent the last three months performing new experiments with newly designed constructs and reagent, reproducing and confirming previous results and reanalyzing data according to the reviewers' detailed suggestions. As a result, our results are greatly strengthened as you will see in the revised manuscript. We really appreciate the amount of time and effort reviewers put into reviewing and commenting on our manuscript and we hope we addressed all the concerns.

## REVIEWER 1

The Myong laboratory studied the transcriptional output of a model gene harboring potential G-quadruplex forming sequences (PQSs) of varying length on the template or non-template strand of the 5`-untranslated region during T7 RNA polymerase-mediated transcription. They found the presence of a PQS on the non-template strand led to more efficient transcription, regardless of length, than the case with the variable PQSs on the template strand. The authors conclude an R loop structure that forms when the PQS is on the non-template strand is responsible for the effect observed.

One set of control experiments should be conducted to support the R loop. Transcription efficiencies should be conducted with inosine triphosphate (R loop destabilizer) or 7-deazaguanosine triphosphate (G4 destabilizer for the transcript) to support the R loop claim and the role of a G4 on the mRNA. A recipe for these experiments was previously reported by the Hanawalt lab (PNAS, 2010, 107, 12816). Indeed, the Hanawalt publication should be referenced in any case since it leads to similar conclusions.

- ➢ We truly appreciate this suggestion. The experiments performed with inosine triphosphate (ITP, R loop destabilizer) led to one key result which helped explain and clarify our findings. Below are four additional figures added to the revised manuscript.

- ➢ We show in **Fig. 7g** (below) that adding ITP in place of GTP resulted in a complete disappearance of the upshifted R-looped bands, which confirmed that ITP indeed removes R-loop by destabilizing it. This also confirms that the upshifted band is the R-loop.

- In addition, we show in **Supplementary Fig. 8b** (below), that in the presence of ITP, R-loop peak in smFRET experiment for both smFRET2 (purple) and smFRET3 (orange and light blue) completely disappear, leaving behind the peaks corresponding to duplex DNA (green in both).

b

- To test the effect of ITP in transcription of PQS-NT, Control and PQS-T, we performed EMSA analysis (**Supplementary Figure 9**, shown below). The R-loop in red and RNA stained in green were quantified and plotted in Figure 8c (the next figure below).

- We quantified the fraction of G4 by the smFRET 3 construct, fraction of R-loop by EMSA band (red, above) and the mRNA yield from EMSA gel (green, above) are plotted in the graph shown below. We demonstrate the striking impact of ITP which completely removes both R-loop, G4 and abolishes the differences in transcriptional output of PQS-NT, PQS-T and non-PQS control which is represented in the new **Fig. 8c** shown below.

- Together, the new data taken in the presence of ITP clearly indicate that:
 - (i) Upshifted gel band on EMSA represents R-loop
 - (ii) Additional peaks in smFRET indicate R-loop structure that forms during transcription
 - (iii) Removing R-loop abolishes G4 formation in PQS-NT.
 - (iv) Removing R-loop reduces differences in transcription level regardless of PQS orientations.

We added the following sentences to address the ITP/R-loop removal effect.

“To directly test the role of R-loop in G4 formation and transcription, GTP was substituted by inosine triphosphate (ITP), which inhibits R-loop formation since inosine forms less stable base-pair with cytosine (Fig 7g). The EMSA gel showed a complete disappearance of the R-loop band (up-shifted) bands under ITP condition, indicating that the up-shifted band is indeed the R-loop structure. In single molecule experiments (Supplementary Fig. 8b) conducted in ITP condition, the 0.1 peak in FRET2 and 0.4, 0.9 peaks in FRET3 all disappeared, reflecting that without R-loop, G4 structure cannot form. This is consistent with our observation that G4 folding depends on the R-loop formation.” (page 10, last paragraph)

- We included the recommended references. (We had them earlier, but had to remove them due to the required length limit).

Various aspects of the introduction and conclusions are confusing, including some ambiguity about whether this manuscript and the papers referenced are talking about G4s in untranscribed promoters or in 5'UTRs. For example, G4s in promoters have been studied in various locations to give increases in gene expression from both strands (Fleming et al, NAR, 2019). Also, the work of Fleming and Burrows who studied strand dependence (up and down, correlating with the work here) was not referenced at all, and they have several papers on this starting with PNAS in 2017, and more specifically for up/down strand dependence, in ACS Chem. Biol. 2017 in the case of oxidative damage to the PQS.

- We apologize for the points of confusion. We addressed the points raised by the reviewer by including all suggested references and clarifying the information. Please see the changed text marked in red in the introduction of the revised manuscript.

More labeling is needed for readers to decipher the figures. What are the red strands? What are the orange-filled duplex regions? What do the gray arrows indicate? The figure captions need to spell out these points that are only sometimes labeled.

- We thank the reviewer for this comment. The revised figures and figure captions have been updated with more clear labels.

I think there is interesting data here, but it needs to have the English polished a little bit and then to make stronger conclusions regarding the impact of a folded G4. For example, the changes seen for template vs. non-template effects appear to be small, if I am correctly reading Figs. 1e, 4d and 5c.

- We prepared a new supplementary table in which we provide numerical data.
- The smaller difference seen in reporter-based assays is due to the limited concentration of the reporter probe. We addressed this by adding the sentence below in the revised manuscript.

“We note that the reporter probe concentration was limited for obtaining reproducible transcription readout without increasing too much background. Therefore, the differences in transcriptional output may be underestimated compared to the other results based on EMSA and single molecule assays presented below.” (page 4, 2nd paragraph)

- To address the G4 induced impact on transcription, we carried out series of additional experiments including EMSA and smFRET as shown in the revised manuscript. (see below)

“Stable G4/R-loop structure enhances RNA production

So far, we have demonstrated that transcription by RNAP on PQS-NT induces R-loop formation, which in turn leads to G4 formation in NT strand. Accordingly, removal of R-loop by ITP incorporation led to disappearance of G4. Next, we asked how the R-loop and G4 structures contribute to RNA production. To modulate the degree of G4 formation, we varied buffer conditions from G4 destabilizing to G4 stabilizing conditions in the order of no monovalent cation, LiCl, KCl, NMM (G4 stabilizing ligand) and pre-formed G4. First, we quantified the fraction of G4 by FRET3 histogram, which displayed the expected pattern of increasing G4 formation as a function of G4 stabilizing conditions (Fig. 8a). Next, we performed EMSA based transcription assay which allowed for measurement of R-loop and RNA production in varying buffer conditions (Supplementary Fig. 9). We plotted the amount of RNA as a function of R-loop and G4 in the form of a bubble graph in which the bubble size corresponds to the level of transcription (Fig. 8b). The graph shows that (i) the R-loop and G4 formation are correlated for PQS-NT, (ii) RNA production of PQS-NT is correlated with both G4 and R-loop; (iii) RNA production of PQS-T is not impacted by G4 or R-loop, (iv) All differences completely disappear with ITP treatment, which abolish R-loop formation (Fig. 8c).

In summary, 5'-UTR PQS influences transcription level through forming R-loop which leads G4 folding that stabilizes R-loop (Fig. 8d). All the transcription processes can potentially

induce R-loop as shown by non-PQS control (Fig. 8d). Both PQS-NT and PQS-T induce G4 formation on NT and T, respectively, yet with opposite consequences. The G4 on PQS-NT stabilizes R-loop by G4/R-loop structure and produces more RNA; by contrast, the G4 in PQS-T blocks transcription and reduce the overall RNA production. Such contrast in orientation dependent G4 effect was accentuated when G4 was pre-formed. The pre-formed G4 in PQS-NT yielded the highest amount of RNA whereas pre-formed G4 in PQS-T produced no RNA (Fig. 8b).” (page 11)

Reviewer 2

Lee et al. have performed in vitro single molecule and bulk measurements on constructs that contain a G-quadruplex (G4) forming sequence in either the template or non-template strand of a double stranded DNA (dsDNA) that also contains an upstream promoter site for T7 RNA polymerase. They have studied the impact of having such potentially G4 forming sequences in the template (PQS-T) or nontemplate (PQS-NT) strand on transcription and R-loop formation. They propose that PQS-NT increases transcription production rate and yield. They attribute this increase to R-loop formation, which induces G4 in the non-template strand. This G4 is proposed to enable successive rounds of R-loop formation and increase transcription. The authors have performed single molecule FRET, protein-induced fluorescence enhancement (PIFE) and gel electrophoresis measurements to support their conclusions on this important topic. Despite a number of bulk in vitro and in vivo studies on the topic, there has not been a single molecule work that investigates this question (to the best of my knowledge), which is a novel aspect of the study.

Prof. Myong and her lab have extensive experience on G4 structures and both the single molecule and bulk methods used in this study. As the authors also cited, the impact of G4 forming sequences on R-loop formation has been demonstrated as early as 2004 using electron microscopy (Duquette....Maizels, Genes and Dev, 2004). Since then there has been a number of studies on this topic, including some whole genome studies that demonstrated high degrees of correlation between R-loop forming and G4-forming regions. In another recent study (also cited by authors) extensive in vivo studies were performed in different cells to investigate connection with G4 stabilization by ligands and R-loop formation (De Magis et al. PNAS 2019). In particular, the impact of non-template G4 structures on R-loop formation and the resulting DNA damage has been well-established. There are also studies that connect G4 structures within the template strand to transcription inhibition. Given this history, the manuscript by Lee et al. should be evaluated based on its contributions to our mechanistic understanding, which has been identified as the primary goal by the authors as well. My main criticism about the manuscript is that I do not find the proposed mechanism and link between G4 and transcription regulation convincing. I found the connection between G4 structures and R-loop formation to be better justified, but in my opinion, these are the already better known aspects of the field. My criticisms are as follows:

Major Criticisms:

1. The authors claim that their studies demonstrate diminished transcription when PQS is in the template strand. However, I do not think the assays that probe transcription show a significant difference between PQS-T and the Control cases in terms of the number of elongation events (Fig 4d), transcription rate (Fig 5c) or rate of Cy3 burst (Fig 1h), which is proposed to depend on the amount of transcribed RNA. Based on visual estimate of the error bars (I did not see them explicitly given), the following numbers are reported: Fig. 1h: Rate for Control 1.3 ± 0.25 ; Rate for Template 1.2 ± 0.2 Figure 4d: % successful elongation for Control 50 ± 5 ; % for Template 45 ± 2 Figure 5c: Rate for elongation events for Control 3.4 ± 0.4 ; for Template 3.2 ± 0.2 None of these numbers are particularly conclusive in demonstrating a diminished transcription. The differences are more prominent when levels of R-loop formation are compared (0.1 FRET peak population), but this does not mean diminished transcription. So, the claim for diminished transcription should be justified.

- We thank reviewer for these points. Based on our results, we see a greater difference in transcription impacted by PQS-NT rather than PQS-T. As we clarify in the revised

manuscript, R-loop is the main player in setting this difference. PQS-NT induces R-loop, which in turn allows G4 formation. In this R-loop/G4 context, transcription output is increased by a mechanism of successive R-loop displacement/reformation.

- Having said above, we note that there are some limitations in certain assays that lead to underestimation of transcription event estimation.
 - (i) Ensemble and single molecule assay included in Figure 1 underestimate transcription events because reporter probe concentration had to be limited due to high fluorescence background.
 - (ii) smFRET elongation assay (Figure 4, dual-PIFE) likely underestimates the elongation frequency because movies are taken only for 2-3 minutes per time, which may be too short for catching slower events. Due to dual excitation of both fluorophores, the photobleaching occurs more frequently, which limits the traces that we can use to analyze this parameter.

- Therefore, we used the following three assays which are not prone to underestimation to assess the transcription output amongst PQS-T, Control, PQS-NT. These measurements were repeated and reproduced more than five times for each conditions.
 - (i) EMSA analysis which enables quantitation of R-loop and mRNA product. Since the reaction is done in the standard condition and the entire reaction is run on the gel, there is no inherent limitation in this method.
 - (ii) FRET histograms generated by FRET3 are used for measuring G4 formation. FRET histograms are collection of 3000-4000 molecules which are measured before the molecules are photobleached. Therefore, the method does not have an inherent limitation.
 - (iii) FRET histograms generated by FRET2 and 3 are both useful for R-loop measurement since distinct FRET state is produced by formation of R-loop and G4/R-loop states.

- We included statistical values corresponding to our measurement in the new supplementary table shown below.

Supplementary Table 3: Statistics results

Table 3.1: Normalized rate of Cy3 increase (Fig. 1e)

Name	Rate	Standard error
Control	1.00	0.01
111-T	0.81	0.01
cMyc-T	0.88	0.02
144-T	0.97	0.01
222-T	1.02	0.01
199-T	0.81	0.02
333-T	0.80	0.05
TTA-T	0.85	0.01
555-T	0.91	0.01

AAA-T	0.92	0.03
111-NT	1.09	0.01
cMyc-NT	1.20	0.01
144-NT	1.22	0.01
222-NT	1.22	0.01
199-NT	1.24	0.01
333-NT	1.07	0.01
TTA-NT	1.16	0.01
555-NT	1.23	0.01
AAA-NT	1.22	0.01

* Standard error is presented as 95% confidence limit.

Table 3.2: Cy3 burst rate (events per min) (Fig. 1h)

Name	Average rate	Standard error	N value
Control	1.32	0.24	164
Template	1.20	0.18	222
Non-template	1.81	0.16	475

* The number was fitted by 1st order exponential decay function.

* Standard error is presented as 95% confidence limit.

- The RNA production and R-looped fraction was directly quantified from the gel as shown in **Supplementary Fig. 7 & 9** (see below). PQS-T produces less RNA than both PQS-NT and non-PQS control. In fact, RNA production of PQS-T is about 10~20% lower than control.

Supplementary Figure 7

Supplementary Figure 9

- For the revised manuscript, we added new data that was conducted under ITP (inosine triphosphate) substitution of GTP. This replacement can prevent R-loop formation (**Fig. 7g** and **Supplementary Fig. 8b**). We also applied different buffer condition to stabilize or destabilize G4 formation (**Fig. 8**). The result suggests R-loop is correlated to additional G4 formation as well as RNA production. In the other words, we observed lower and non-G4 relevant R-loop formation in PQS-T construct, which shows a steady RNA production no matter the variant of buffer. This can be explained the additional G4 formation in PQS-NT is essential for regulating transcription.

Figure 7g

g

Supplementary Figure 8b

b

Figure 8a, b, c

- In fact, G4 formation on template strand has to compete with the R-loop formation or reannealing back to dsDNA. That is, we think the slightly lower RNA production is due to the G4 formation in template strand, which is indistinguishable in our current smFRET construct.

Also, more prominent reductions in transcription have been reported for other systems when G4 is in the template strand. For example, a 3x enhancement in c-Myc expression was demonstrated when G4 forming sequence was eliminated by a single nucleotide mutation (SiddiquiJain..Hurley, PNAS 2002). Can the authors comment on why the impact is much smaller in their assays?

- We appreciate the reviewer's comment. We think this is due to the differences in study design. In SiddiquiJain and Hurley's paper, the effect was focused on promoter region, whereas our PQS was located downstream of promoter. In Sugimoto's paper (2018 & 2019) as referenced in discussion, they examined the template PQS in transcription, which block the transcription level, which agrees with our finding.
2. In essentially all the data in Figures 1,4, and 5, the rates for Control and PQS-T are very similar (within the error bars), while the rates for PQS-NT constructs are slightly higher than those for PQS-T. In Figures 4 and 5, the rates of PQS-NT and Control are also very similar considering the error bars associated with the measurements. In general, the difference between the quoted rates is marginal. On the other hand, the populations of the 0.1 FRET peak in Figure 5d are very different for these cases. In particular, the population of this peak for the non-template construct is 2-3 times higher than that for the template construct throughout the measurement. This peak is interpreted to represent R-loop formation, which is proposed to be the underlying mechanism for enhanced transcription in PQS-NT constructs compared to PQS-T constructs. The authors have not clarified the reasons for this quantitative discrepancy. If stable R-loop formation gives rise to higher levels of transcription, why isn't there a quantitative agreement between the two observations? One might even take the opposite perspective and propose that despite significantly higher levels of R-loop formation, the rates of transcription differ only marginally.
- We have addressed this points above. Please see our answer for #1. Basically, some methods we used here such as reporter based transcription assay and dual-PIFE based elongation assay tend to underestimate differences due to inherent limitations as explained above (#1). Nevertheless, the reporter based ensemble assay was a useful tool for screening many G4 sequences in template, control and non-template strand. In addition, despite the underestimated differences, all methods displayed consistent pattern of PQS-NT > non-PQS control > PQS-T.
3. I found the evidence provided for successive R-loop formation to be weak and thought the single molecule and gel measurements were not coherent. For example, the authors suggest the spikes observed in Figure 6b to be evidence for successive rounds of R-loop formation. These jumps are ~10s apart and they take place at a constant 0.1 FRET base state. This would require not to have any time gap between one R-loop and the next, which is not clear how it could be possible. Also, in the EMSA it takes many minutes for Cy5-labeled UTP to be displaced by an unlabeled UTP, but in the single molecule traces the time difference is only ~10 s. It is not unexpected that given enough time, the RNA strand within an R-loop will be displaced by another RNA strand in a second round of transcription, which would result in replacement of the labeled UTP with the unlabeled one. But this is different from observing successive R-loop formation events within tens of seconds apart (5 events within ~120 seconds in Figure 6b). Why do the authors rule out simpler explanations such as successive binding of RNAP within the vicinity of Cy3? As shown in the traces in Figure 6b, such jumps take place even at the higher FRET state, before the transition to the 0.1 FRET state.
- We understand the reviewer point about the discrepancy between single molecule trace and the gel data. In **Fig. 6b**, we showed these traces to demonstrate that the 0.1 FRET

represents R-looped state, rather than due to photobleaching and that the transcription was still active in R-looped state. Importantly, the Cy3-PIFE spikes only occurs in transcribing condition i.e. RNAP without NTP never shows any signal spikes, suggesting that they correspond to active transcription by RNAP.

- However, not every RNAP makes it through the full round of transcription, as we demonstrated in dual-PIFE assay. Therefore, the high frequency of spike may be overestimating the rate of successive transcription.
- Based on the slow time scale of Cy5-UTP incorporation as well as removal (Fig. 6c, d) shown both in gel based analysis and single molecule detection likely indicates a significantly slower removal of R-loop that contains Cy5-UTP. We addressed this by the following statement added to the revised manuscript.

“We note that the lower rate of R-loop replacement than the rate expected from the Cy3 spikes seen in FRET2 construct (Fig.5c and 6b), is likely due to the dye labeled UTP in the transcript which cannot be removed by RNAP as efficiently.” (page 8, second paragraph)

Along these lines, I do not think Figure 6d also provides much support for the successive R-loop formation since it is expected that as RNA is synthesized the number of spots in the Cy5 channel will increase (more labeled UTP is incorporated) and when the chamber is treated with RNaseH these spots will be eliminated.

- We agree with the reviewer’s comment. The original purpose of **Fig. 6d (old)** is to show that the Cy5 signal appeared with successful incorporation of Cy5-UTP and disappeared after RNase H removed R-looped RNA. We agree that it does provide evidence about successive R-loop removal/formation, so it is removed from revised manuscript.
4. The molecular beacon in Figure 1 will be complementary not only to RNA but also to the nontemplate strand. So, the rise in Cy3 intensity (Figure 1c) could be due to not only binding of the beacon to RNA but also to the non-template strand. Later in the paper, the authors show that R-loop formation (between RNA and the template strand) is the dominant mechanism during transcription for these GC-rich sequences, particularly when G4 is in the non-template strand (PQS-NT). However, the analysis is performed assuming R-loops do not form, and transcribed RNA is available for binding to the molecular beacon. The authors should provide a justification for neglecting R-loops and their potential impacts on the binding characteristics of the beacon. This is particularly important since R-loops are not equally likely to form or remain stable for PQS-T and PQS-NT samples, and therefore, will impact the two constructs at different levels. The same criticism is valid for the gel analysis as well. If R-loop formation impacts the availability of the RNA to hybridize with the beacon, then the beacon intensity does not necessarily correlate with the amount of transcribed RNA. These effects are complicated to decouple since it is not clear which segment of RNA takes part in R-loop formation, i.e. it is not known whether the segment that is complementary to the beacon takes part in R-loop formation.
- We appreciate the reviewer’s comments. In our gel based transcription analysis (Fig. 1b), we do notice that the beacon binds DNA template; however, as RNA continues to be produced, the Cy3 signal in DNA subsides while Cy3 signal on RNA becomes brighter as a

function of transcription time. This indicates that there is a competitive binding of the beacon between the DNA (both duplex and R-loop) initially, but the beacon-RNA takes over once the RNA concentration overwhelms the DNA template concentration of 1nM. As we stated in the manuscript, the beacon concentration is 100nM and DNA template concentration is 1nM.

- In addition, the RNA production was quantified by not only the beacon intensity but also by the RNA staining dye (UV illumination) on the EMSA gel, both of which produced the comparable differences with the same pattern of PQS-NT > PQS-T.
- For single molecule beacon assay, we have done series of control experiments to make sure that we select correct binding signal. That is, if the beacon bound non-template, the signal will be long lasting (few minutes until photobleaching), since the DNA is tethered to surface. However, all the beacon signal is extremely short-lived, occurs in short burst in all cases. In fact, we rarely observed R-loop binding on non-template, which maybe because the DNA concentration (less than 25 fM) is too low to reach equilibrium state, and it also indicates that the probe preferentially binds mRNA and is released into solution.
- More importantly, as stated above (#1), the beacon assays were used only as a screening tool since it has inherent limitations due to limited concentration used in the assay to avoid background fluorescence. As stated, all the quantitation regarding R-loop formation, G4 formation and mRNA production was performed by more accurate, unbiased assays including FRET2, FRET3 histograms and EMSA analysis.

Minor Criticisms:

1. The authors seem to suggest R-loop formation gives rise to G4 formation in the non-template strand, which in turn stabilizes the R-loop. Whether R-loop or G4 comes first is not clear, and I am not sure if there is any evidence in the manuscript supporting the proposal that R-loop formation comes before G4. Based on crystal structure, the RNA transcript and the template strand of DNA are known to come out of different channels of the RNA polymerase. This RNA transcript likely folds back on to the template strand to form the R-loop. However, during transcription and while the transcription bubble is open, the non-template strand should be relatively free to form G4, which makes the possibility of G4 forming before the R-loop to be potentially possible.
 - We apologize that this information was less than clear. We did demonstrate that R-loop forms before G4. In the third trace shown below (**Fig. 7d**), the green (DNA-only) state converts to orange (R-loop), then to light blue (G4). The second trace demonstrates a case in which duplex DNA goes to R-loop then back to duplex etc. Once the R-loop converts to G4 (R-loop/G4), it never comes back down to R-looped state. We clarified this in the manuscript as shown below.

“Furthermore, the smFRET3 (PQS-NT) trajectories reveal several mechanistic details (Fig. 7d). First, the dynamic exchange between 0.25 and 0.4 FRET states indicates that the R-loop can form transiently and can return back to the duplexed form. Second, 0.9 state is only reached after 0.4, indicating that the R-loop is required to form G4. Third, the steady 0.9 state reveals that the G4/R-loop state is a stable conformation which does not return to the other two states. The high FRET state remains after RNase H digest in FRET2 and FRET3, respectively, indicating that the G4 persist even after the R-loop is removed. In order to distinguish the G4 from other DNA, we ran a potassium PAGE gel (10% PAGE with 100 mM KCl). The up-shifted R-loop band, after the RNase H treatment, yields the linear DNA band and an additional band at the position of the pre-formed G4 DNA, confirming the G4 formation during transcription, which is stable even after R-loop removal (Fig. 7e).“ (page 9-10)

2. In Lines 46-48, the authors wrote: “Furthermore, there is no direct evidence to attribute the observed transcriptional effect of PQS to G4 formation rather than the guanine rich sequence itself.”. I don’t think this statement is accurate since there are studies that directly probed the impact of G4 by mutating the sequence (including single nucleotide mutations) and demonstrated significant change in both mRNA and protein levels. The study from Hurley’s lab, which was mentioned before, is an example for this (Siddiqui-Jain..Hurley, PNAS 2002).
- We apologize for making this statement. We have added more references including the suggested references and restated the sentence as shown below.

“Furthermore, although previous studies have shown correlation between PQS at 5’-UTR and transcription level, there is no direct evidence to show if G4 actually forms during transcription reaction.” (page 2, second paragraph)

3. Lines 97-98: Why does Cy3 show up as spikes? Is Cy3 photobleaching or dissociating from its complement? How do the authors distinguish Cy3 molecules that non-specifically bind to the surface from those that bind to the complement? 100 nM is much higher than typically used for donor labeled molecules in smFRET studies, but having a quencher has enabled performing the studies without excessive background. Nevertheless, this high concentration would result in significant transient nonspecific binding to the surface, even a high-quality PEG surface. Even though the quencher suppresses Cy3 signal while the beacon is in the solution and flexible, it may not do so when it the beacon is transiently bound to the surface. In this

immobile state, the end-to-end separation could be significant resulting in imperfect quenching. Have the authors performed a similar assay in an empty channel to estimate the level of such non-specific transient binding events?

- We have done series of control experiments to distinguish and select the correct Cy3 signal. All non-specific binding or photobleaching can be filtered out by dwell time thresholding analysis.
 - The design of quencher was introduced by IDT company, and method protocol was adopted from a previous publication. Please see the following paper in which they used beacon in single molecule transcription assay.
- Zhang, Z., Revyakin, A., Grimm, J.B., Lavis, L.D., Tjian, R. "Single-molecular tracking of the transcription cycle by sub-second RNA detection", *Elife*, 2014;3:e01775.
4. In Figure 4c: What criteria did the authors use while counting the PIFE events? Even in these very nice traces, there are some events that may or may not be counted as PIFE events. For example, there two events around time=50s in the top trace of Figure 4c which are clear spikes in the in the Cy3 signal but are not as prominent as those shown with green arrows. There are similar events in other traces as well, which would require many judgment calls unless well-defined selection criteria are established.
- Please see our answer to #1 and #4 above. In addition, we have carefully carried out series of negative and positive experiments to distinguish PIFE events corresponding to the true transcription event.
 - PIFE has been extensively used in our research and we have a stringent criterion for distinguishing PIFE signal from random signal fluctuations. We perform a series of negative and positive controls. We calibrate PIFE signal for each protein-nucleic acid system before taking any measurement. Please see the following papers in which we used PIFE for various protein-DNA, RNA systems.
- Niaki, A.G.* , Sarkar, J.* , Cai, X., Rhine, K., Vidaurre, V., Guy, B., Hurst, M., Lee, J.C., Koh, H.R., Guo, L., Fare, C.M., Shorter, J., Myong, S. "Loss of dynamic RNA interaction and aberrant phase separation induced by two distinct types of ALS/FTD-linked FUS mutations", *Molecular Cell* 2019, Vol. 77, Issue 1, p82–94.e4
 - Koh H., Kidwell M., Doudna J., Myong S. "RNA scanning of a molecular machine with a built-in ruler", *Journal of the American Chemical Society*, 2017, 139 (1), pp 262–268.
 - Qiu Y., Myong S., "Chapter Two - Single-Molecule Imaging With One Color Fluorescence", In: Maria Spies and Yann R. Chemla, Editor(s), *Methods in Enzymology*, Academic Press, 2016, Volume 581, Pages 33-51.
 - Wang X., Vukovic L., Koh H., Schulten K., Myong S., "Dynamic profiling of double-stranded RNA binding proteins", *Nucleic Acid Research* 2015 Sep 3;43(15):7566-76.
 - Koh H., Xing L., Kleiman L., Myong S., "Repetitive RNA unwinding by RNA helicase A facilitates RNA annealing" *Nucleic Acid Research* 2014 Jul;42(13):8556-64.

- Hwang H., and Myong S. "Protein induced fluorescence enhancement (PIFE) for probing protein–nucleic acid interactions" *Chem. Soc. Rev.*, 2014 Feb 21;43(4):1221-9.
 - Qiu Y., Antony E., Doganay S., Koh H., Lohman T. M., and Myong S. "Srs2 prevents Rad51 filament formation by repetitive motion on DNA". *Nat Commun.* 2013 Aug 13;4:2281.
 - Hwang H., Kim H. and Myong S. "Protein induced fluorescence enhancement as a single molecule assay with short distance sensitivity" *PNAS* 108(18):7414-8 (2011)
 - Myong, S., Cui S., Cornish P. V., Kirchhofer A. , Gack M. U., Jung J. U., Hopfner K. P. and Ha T., "Cytosolic viral sensor RIG-I is a 5 prime - triphosphate-dependent translocase on double-stranded RNA", *Science* 323(5917), 1070-1074 (2009)
5. Lines 150-153: the authors wrote: "Approximately 50% successful elongation was displayed in nonGQ control whereas 45% and 58% was observed for PQS-T and PQS-NT, respectively (Fig. 4d), strongly suggesting the higher rate of successful elongation in PQS-NT than the non-PQS and the PQS-T." Given the error bars (~5% in the control), the proximity of the cited averages, and the complex nature of the data analysis regarding selection criteria, I think "strongly suggesting" in this statement is not justified.
- As stated in #4 above, the dual-PIFE method is prone to underestimate. Therefore, other methods were used to quantify the differences in R-loop, G4 and transcription output.
 - For the revised data, we have carefully re-processed our data with accurate statistical method. Here is the numerical data, showing in new **Supplementary Table 3**.

Table 3.7: Fraction of successful elongation (Fig. 4d)

Name	Average	Standard error	N value
Control	52%	2%	1839
Template	44%	3%	997
Non-template	58%	2%	1585

* Standard error is determined by binomial distribution with 95% confidence interval.

6. Lines 162-163, Figure 5d, and Fig 6b: In Figure 6b, the authors have shown the R-loop construct as linear while the dsDNA itself is in the form of a double helix. I understand that this is just a schematic, but it could be misleading since the R-loop will also have the double helix structure, probably close to the A-form. Since the A-form has shorter separation between the base pairs (although with a little greater diameter), one would expect a DNA-RNA hybrid to have a shorter end-to-end distance compared to the standard B-form dsDNA. However, the authors have proposed the R-loop to have FRET 0.1 while the dsDNA has FRET 0.3 for the same dye-positions. Can the authors elaborate on why the Rloop seems to have greater end-to-end distance compared to dsDNA? Does RNAP binding to an unwound dsDNA contribute to this lower FRET? Have the authors measured the FRET level that corresponds to RNAP binding to a transcription bubble that has Cy3-Cy5 at similar bp separation to their construct?

- We understand the concern. FRET does not only depend on the length but also on the dipole orientation of the dyes (Asif Iqbal et al. PNAS 2008). In our smFRET2 construct, the two dyes are labeled on different strands, so it cannot be interpreted as end-to-end distance. Therefore, we cannot assign particular structures to each case. To address this issue, we prepared a summary chart shown below (**Supplementary Fig. 8a**) to demonstrate all potential structures and FRET arrangement.

7. Lines 168-169: If the dsDNA is 90-bp, why does the R-loop run at 500-bp in the gel (Figure 5f)? Is this expected or reported before? Is RNAP expected to remain bound to the promoter on the dsDNA during this gel? It looks like for the RNaseH treatment, the RNAP is removed from the construct by adding high concentration of the T7 promoter before adding RNaseH. But it is not clear if RNAP is removed for the measurements before RNaseH. This question stems from the ambiguity about whether the shifted band at 500 bp could be due to binding of RNAP to the promoter on the DNA and not necessarily only due to the R-loop. A clarification of this would be helpful.

- We did rule out this possibility by adding 0.1% SDS into the samples before loading to the gel. It's already known that 0.1% SDS is enough to disengage protein from substrate and we have confirmed in our single molecule assays of this and other unrelated cases. Therefore, the shift most likely reflect DNA. We added this detail in the revised manuscript as shown below.

“The samples were treated with 0.1% SDS to denature proteins and eliminate RNAP-DNA complex.” (page 7, first paragraph)

8. Line 191-192: the transition from 0.3 to 0.1 FRET is very fast as far as I can tell from the traces, possibly as fast as the time resolution of the measurements. Do the authors expect the proposed dsDNA to R-loop transition be this fast? The dual-PIFE measurements suggest that there are several second time difference between RNAP interacting with Cy3 and Cy5, which should set the lower time scale for the transition from duplex to R-loop, e.g. bubble creation, RNA transcription, and invasion of the DNA by RNA should take place before duplex to R-loop transition takes place. The other possibility is that the Rloop forms after the RNA is already transcribed and duplex reanneals. However, in that case it is interesting that none of the other dynamics mentioned above shows up in the traces and only a stable 0.3 FRET state is observed before R-loop formation is observed.

➤ This sudden transition likely reflects the transcription bubble opening induced by RNAP rather than the R-loop formation. Upon R-loop formation, the 0.1 FRET state appears to be maintained rather than returning back to 0.3 FRET.

9. Line 223: Is the population of the 0.9 FRET peak only 20% of the total population at 60 min in Figure 7c? It appears to be much higher. Especially considering the population of the 0.9 FRET peak in Figure 5e is also quoted to be 20% of the population. The peak in Figure 7c appears to be 2-3x more populated than that in Figure 5e.

➤ This is because we only compare the FRET histograms after the RNase H treatment. Indeed, during transcription, there is a higher level of 0.9 FRET peak as shown, but not all of them remains after RNase treatment. The FRET histogram was fitted by multi-Gaussian function, and the percentage number was calculated by the area-under-curve ratio

On a related point, if an R-loop can form with or without the G4 in the nontemplate strand, why isn't there any R-loop with 0.9 FRET in Figure 5d? I understand that the Cy3 is moved from the template to the non-template strand however, this should not create a distance change large enough to reduce FRET 0.9 to 0.1. At least the R-loop peak remains at 0.1 FRET in both cases even though Cy3 is on the template strand in one case but on the non-template strand in the other. The 0.9 FRET peak appears only after RNaseH treatment (Figure 5e), which removes the RNA strand but does not influence the fluorophore positions. So if G4 can form (with or without the R-loop) in the FRET2 construct, it should be at FRET 0.9. Figure 7f mid-panel shows that the 0.9 FRET population persists even after R-loop formation, which suggests it should have appeared in Figure 5d as well. Given these, I am not convinced that the 0.1 FRET peak in Figure 5d represents R-loop with and without G4 formation.

➤ Please see our answer for #6 above. The template and not-template should not be thought of as a contiguous entity, especially when G4 and R-loop structure separate the two strand. Each strand should have its own degree of freedom that controls dye position and orientation such that we obtain lower than expected FRET value.

10. Line 246: The authors wrote: "...the FRET2 entirely shifted to 0.1..." when describing middle panel in Figure 7f. However, a peak at 0.9 FRET is clearly visible in the histogram, so the shift does not seem to be complete.

➤ We have modified the statement accordingly.

Lines 247-249: What is meant by the statement “The results....RNaseH digestion.” is not clear to me.

- To avoid confusion, we rewrote the sentences as follows.

“Upon RNAP reaction, the FRET2 shifted to 0.1 and shifted back to 0.9 after RNase H treatment, indicating that the 0.1 state prior to RNase H treatment was G4/R-loop state. FRET3 peak remained at high FRET in both conditions (Supplementary Fig. 5a), indicating that the pre-formed G4 in NT remains stably folded, unperturbed by transcription and RNase H digestion.” (page 10, second paragraph)

Line 385: Is Trolox at ~10 mM concentration? Many labs quote ~2mM as saturating concentration in physiological pH buffers.

- This is the stock concentration which ranges from 2 mM to 10 mM.

Figure 6e: In such analysis, typically the control for photobleaching is also demonstrated.

- This analysis was done by a quick (2 sec) imaging of many different areas for the purpose of collecting as many molecules as possible without photobleaching. We note that this figure is now Figure 6d in the revised manuscript.

d

Single molecule RNA replacement assay

Figure 7f: It is surprising that no 0.4 FRET peak is observed in the Pre-formed G4 case for FRET3 construct after +RNAP+NTP are added. Given that state dominates in Figure 7c (when G4 is not preformed), I would have expected at least some population at that peak. Since effectively nothing has changed in the histograms for FRET3 construct after +RNAP+NTP or RNaseH treatment, do the authors know whether transcription took place in these FRET assays? As the R-loop readily forms and is very stable in all other assays, it is surprising that it does not form at all in this assay.

- G4 folding is extremely stable both in pre-folded state and when formed in the process of transcription. In FRET3, once 0.4 (R-loop) transitions to 0.9 (G4/R-loop), the 0.9 state never

returns to 0.4. We are confident that transcription took place as shown below (**Supplementary Figure 7**, shown below) by the green RNA bands that clearly increases over time while yellow G4/R-loop is maintained.

Typos: Line 55 (incomplete sentence), Line 58 (has/have), Line 62 (each stage), Line 123 (has/have), Line 143 (which puts), Line 168 (at 10-minute intervals), Line 610 (transcribed RNA), Line 611 (up traces?), Line 612 (as shown in f), Line 613 (as shown in f).

➤ We thank the reviewers for pointing them out. The typos are corrected.

Reviewer #3

The work by Lee et al. assesses the potential role of G-quadruplex (G4) in promoting transcription. Using a combination of bulk and single-molecule experiments the authors show that the presence of a G4-forming sequence downstream a TSS and on the non-template strand stimulates transcription. The main conclusion of the manuscript is that while the presence of the G4-forming sequence does not affect T7 RNA binding and transcription initiation, it affects the elongation step. The authors propose that G4 formation in the non-template strand favour R-loop formation that in turn stimulates transcription elongation. Because the exact mechanism by which G4 can promote or repress transcription are not well characterised, this manuscript is of potential interest for the broad readership of Nature Communication. Nevertheless some of the conclusions of the manuscript are not fully supported by the data and additional control experiments are needed to support the role of G4 in R-loop stabilisation. Here are some comments/suggestions:

Main points:

(1) The data reported in the current manuscript support that a guanine-rich sequence in a non-template DNA strand promotes transcription, which is in stark contrast with previous reports (Belotserkovskii et al. PNAS 2010, NAR 2012 and NAR 2017). The experimental setups in these works differ only by the position of the guanine-rich sequence compared to the TSS. Hence the position of the guanine-rich sequence seems crucial in determining its inhibitory or stimulatory effect. Because the novelty of the current manuscript lies in the stimulatory effect of the guanine-rich sequence it seems essential to study the positional effect of the guanine-rich sequence.

- We thank the reviewer for this comment. We do know that position makes a difference. Below we summarize what we have done.
- We tried total of 4 positions including the previously employed **9 bp** (Belotserkovskii et al. PNAS 2010), **16 bp** (Koh HR et al. Mol Cell 2018), **31 bp** and **41 bp** downstream of the promoter. For 16 bp, 31 bp and 41 bp, we have tested more than 9 different PQS, and they all showed the same pattern of higher RNA production in PQS-NT than PQS-T. For 9 bp construct, however, the difference nearly disappeared.
- We used 16 bp for smFRET experiment and EMSA and obtained the same pattern of difference i.e higher transcription in PQS-NT than PQS-T (**Fig. 8b**, and **Supplementary Fig. 6, 9**). To address the discrepancy with other studies, we included the following statement in discussion section.

“Previous studies reported both up- or down regulation of transcription and translation led by different PQS orientations and such conflicting results may arise from differences in experimental design including position of PQS with respect to promoter, distance from the transcription start site, length and composition of G-rich sequence^{7, 8, 20, 27, 49}. How position, distance, extent and composition of PQS influences transcription and translation will be studied further in the next phase of our study.” (page 13-14)

(2) The main conclusion of the article is that G4 folding within the non-template strand stabilise

the R-loop. Nevertheless the authors did not demonstrate accurately this point and the data reported in Fig 1e suggest otherwise (e.g. the AAA-NT sequence which is not prone to G4 folding according to the NMM-based fluorescence assay stimulates transcription). To support the conclusion of the paper, the authors need to demonstrate that the structure on the non-template strand rather the guanine-richness of the synthesised RNA stabilises the R-loop. The authors should perform the different transcription assays in the presence or the absence of KCl, using a control sequence of similar G-richness (e. g. point mutation in the c-myc sequence) or in the presence of a small molecules known to stabilise G4s to validate this point. These experiments are crucial since the inhibitory effect of R-loops reported by Belotserkovskii et al. is independent from G4 formation.

- We thank the reviewer for this great suggestion. We have conducted new set of experiments following the buffer conditions suggested by the reviewer. We include this as a new result as part of **Figure 8** in which we summarized the correlations among G4 formation, R-loop formation and RNA production. Below is the text inserted into the revised manuscript and the new Figure 8.

Stable G4/R-loop structure enhances RNA production

So far, we have demonstrated that transcription by RNAP on PQS-NT induces R-loop formation, which in turn leads to G4 formation in NT strand. Accordingly, removal of R-loop by ITP incorporation led to disappearance of G4. Next, we asked how the R-loop and G4 structures contribute to RNA production. To modulate the degree of G4 formation, we varied buffer conditions from G4 destabilizing to G4 stabilizing conditions in the order of no monovalent cation, LiCl, KCl, NMM (G4 stabilizing ligand) and pre-formed G4. First, we quantified the fraction of G4 by FRET3 histogram, which displayed the expected pattern of increasing G4 formation as a function of G4 stabilizing conditions (Fig. 8a). Next, we performed EMSA based transcription assay which allowed for measurement of R-loop and RNA production in varying buffer conditions (Supplementary Fig. 9). We plotted the amount of RNA as a function of R-loop and G4 in the form of a bubble graph in which the bubble size corresponds to the level of transcription (Fig. 8b). The graph shows that (i) the R-loop and G4 formation are correlated for PQS-NT, (ii) RNA production of PQS-NT is correlated with both G4 and R-loop; (iii) RNA production of PQS-T is not impacted by G4 or R-loop, (iv) All differences completely disappear with ITP treatment, which abolish R-loop formation (Fig. 8c).

In summary, 5'-UTR PQS influences transcription level through forming R-loop which leads G4 folding that stabilizes R-loop (Fig. 8d). All the transcription processes can potentially induce R-loop as shown by non-PQS control (Fig. 8d). Both PQS-NT and PQS-T induce G4 formation on NT and T, respectively, yet with opposite consequences. The G4 on PQS-NT stabilizes R-loop by G4/R-loop structure and produces more RNA; by contrast, the G4 in PQS-T blocks transcription and reduce the overall RNA production. Such contrast in orientation dependent G4 effect was accentuated when G4 was pre-formed. The pre-formed G4 in PQS-NT yielded the highest amount of RNA whereas pre-formed G4 in PQS-T produced no RNA (Fig. 8b).

Figure 8 (next page)

(3) The author concludes that the presence of the G4-forming sequence does not affect transcription initiation. Nevertheless the data reported in Supplementary Fig 2e shows that the presence of the G4-forming sequence decrease by ~ 5% the number of abortive events. Knowing that the G4-forming sequence in the non-template strand increases by ~ 8% the number of successful elongation events (Fig 4d), why is the effect of on abortive events ruled out to explain the stimulatory effect of G4-forming sequences?

- We thank the reviewer for pointing this out. We did not state the abortive initiation as difference making step since the differences between conditions were not as significant as other contributors such as R-loop and G4. We will continue to examine this as we move to more complex G-rich sequences in the next phases of our work.
- We also provide numerical data in new **Supplementary Table 3**. Here are the data of the dual PIFE and non-abortive analysis.

Table 3.7: Fraction of successful elongation (Fig. 4d)

Name	Average	Standard error	N value
Control	52%	2%	1839
Template	44%	3%	997
Non-template	58%	2%	1585

* Standard error is determined by binomial distribution with 95% confidence interval.

Table 3.14: Fraction of non-abortive events (Supplementary Fig. 2e)

Name	Average	Standard error	N value
Control	78%	3%	543
Template	81%	3%	589
Non-template	83%	3%	551

* Standard error is determined by binomial distribution with 95% confidence interval.

(4) It is surprising that that the stimulatory effect of the R-loop is observed only for high concentrations of the RNA polymerase. One explanation could be that a high concentration of RNA polymerase is needed to displace the R-loop after the first round of transcription. The author could assess the impact of RNA polymerase concentration on the rate of R-loop displacement using their single-molecule RNA replacement assay reported in Fig 6.

- We thank the reviewer for this insightful comment. We performed RNAP titration assay and the new data (Fig. 5h,i) and text are included in the revised manuscript.

h

i

“The extent of R-loop formation was correlated with RNAP concentration with the most prominent impact in the PQS-NT (Fig. 5h, i).” (page 7, second paragraph)

Minor points:

(5) In order to appreciate the relevance of the differences between the G4-forming sequence in the template, non-template and control constructs (Fig 3d and h, Fig 4d, Fig5c and corresponding supporting figures) the authors should use appropriate statistical tests and report their values.

➤ The statistical result was provided in new **Supplementary Table 3**.

(6) On figure Fig 1d the initial slope of the control sequence (black curve) seems to be bigger than the slopes for the non-template constructs which is not the case on the quantification reported in Fig 1e.

➤ The curve in the original Fig. 1d was taken from one of many experiments, but the quantified data in Fig. 1e was averaged from more than 5 repeated experiments. Also, in order to show the control curve, we specifically thickened control curve, which is fixed in revised version as shown in the right.

Reviewers' comments:

Reviewer #1 (Remarks to the Author):

The authors have addressed my concerns.

Reviewer #2 (Remarks to the Author):

The authors have done a respectable amount of additional work to address reviewer comments, including acquiring additional data and new analysis. They have also made the requested changes in the manuscript to clarify and explain the issues I raised. I am in general satisfied with the new version of the manuscript and explanations of the authors.

There are several minor issues that are either not clear to me or I thought the explanations provided by the authors or their interpretations were speculative. I will list these issues below as these reviews are also made available to the readers. But I do not think these should prevent or delay publication of the manuscript. I leave it to the discretion of the authors whether they consider these worthwhile to address in the final manuscript.

1- The 0.9 FRET peak in Figure 7 is interpreted as the state that has both an R-loop and a G4. However, the only population that we know with certainty to have such high FRET value is the G4 state. I understand that G4 comes after R-loop formation but persists even after the R-loop is removed. So, I believe this population consists both G4 with R-loop and G4 without R-loop. This perspective is also supported by the reduction in the 0.9 FRET peak upon RNaseH treatment (Fig. 7c). The reduction in the 0.9 FRET peak might be due to the loss of G4 with R-loop population and what remains is the G4 without R-loop population. Considering the 0.9 FRET to be R-loop with G4 makes it more convenient to correlate with the data on the other constructs, but I don't think we can conclude that all 0.9 FRET states have an R-loop.

2- I did not find the reasoning and evidence provided by the successive R-loop formation to be convincing, especially when these events are separated by ~ 10 s. I also did not find the explanation for why having an R-loop in place facilitates a new RNAP to start a new transcription event. After all, RNAP or the newly synthesized RNA still has to replace the old RNA that is within the R-loop. This might very well be what is happening, but it remains mechanistically vague for me. Having an R-loop from the previous transcription event might possibly help as it would prevent the template and non-template DNA strands from hybridizing, and this might facilitate the next transcription event.

3- The authors have explained the lower FRET value in R-loop state compared to dsDNA (Fig. 5d) with fluorophore dipole orientation. This is very speculative since no measurement has been performed to probe the dipole orientation in either dsDNA or the R-loop state. For all practical purposes, the orientation factor (K^2) might be greater in the R-loop state compared to the dsDNA state or vice versa. The Iqbal et al. (PNAS 2008) paper which is cited in the response is not a representative paper for the context of this study since in that paper the ends of short duplexes were labeled whereas in this study the fluorophores are internally labeled. It is generally accepted that when placed at the end of a duplex, fluorophores stack on the last base, and are restricted in terms of the orientations they can assume. The situation is quite different when fluorophores are labeled internally. I believe the difference in FRET between the R-loop and the dsDNA is due to displacement of the non-template strand, which results in lower FRET state than the R-loop state.

4- I noticed a couple typos in Figure 3 caption: "300 trances" and "f, As in shown e"

Reviewer #3 (Remarks to the Author):

In the revised version of their manuscript, Lee et al. only partially addressed the comments/issues raised by the reviewers. While the work reported herein is of potential interest, the current manuscript is not ready for publication in Nature Communications or any other journals. The main limitation of the current manuscript lies in the absence of clear evidences demonstrating that G-quadruplex (G4) structure formation in a non-template DNA is necessary for stimulating transcription. The authors did not demonstrate that the observed transcription stimulation is due to the G4 structure in the non-template strand rather than the guanine-rich sequence in the nascent RNA. Because this is the main conclusion of the manuscript and the basis of the title of the manuscript, clear evidences need to be reported. Here are some comments on the revised manuscript:

Main points:

(1) In the most recent report from the Hanawalt lab (NAR 2017), the authors report that the G-richness of a sequence rather than its ability to fold into a G4 structure affect both R-loop formation and transcription. Moreover the authors report that R-loop formation inhibits rather than stimulates transcription. The experimental setup used by the Hanawalt lab only differs from the current study by the position of the quadruplex. Studying the positional effect of the G4-forming sequences is then crucial for understanding the contribution of G4 and R-loop to transcription regulation. Without this information, the current manuscript will confuse, or even worse mislead, the potential readers when trying to understand the role of guanine-rich sequences in transcription regulation. From the answers of the authors to my comments, it seems that they already performed some experiments to investigate this point. While they agree that the position of the G4 affects the outcome of the experiments, their answer remains elusive and they do not provide any experimental data. This is detrimental to their manuscript and these data should be presented.

(2) The authors have now including experiments studying the impact of G4 stabilisation, using KCl or NMM, on transcription. Nevertheless the results presented in the "bubble" graphs (Fig. 8b and 8c) do not seem to agree with the raw data (EMSAs presented in Supplementary Fig. S9). It is unclear how the transcription efficiencies from the different templates and conditions have been quantified. One can see from the gel reported in Supplementary Fig. S9a that the use of the control template leads to the highest transcription efficiencies and that the addition of KCl or NMM do not affect transcriptional outputs. Moreover it seems that an R-loop can also be formed when performing the quadruplex on the template strand. All these observations are in clear contradiction with the data reported in the manuscript (Fig. 8b). If the author's claims are real, they need to report a thorough quantification of transcriptional outputs using replicated data together with a statistical analysis of the differences in transcription. As noted by all reviewers, the use of PQS-T does not show a significant difference from the control template. Taken together these observations challenge the main conclusions of the article.

(3) The authors provide, in the revised version of their manuscript, experiments performed with ITP in order to destabilise the R-loop without affecting G4 formation. However the reported data are either conflicting or incomplete. From the gel reported in Supplementary Fig. 9b, one can see that the use ITP globally inhibit the transcription of all templates as RNA products can barely be seen. This suggests that, under the condition of the assay, ITP is a general inhibitor of transcription. The method for RNA quantification should be reported appropriately, replicated and the relevance of any difference statistically analysed. The authors nevertheless reports some sort of quantification on Fig. 8c, but the data associated with the different conditions (LiCl, KCl, NMM, performed) are missing. These data are the most informative in order to conclude on the role of the G4 motif in the absence of an R-loop. If no differences are observed, one may conclude that the G4 motif does not contribute to transcription regulation but its formation is merely the consequence of the formation of the R-loop.

(4) It is crucial that the authors carefully assess the statistical differences associated to the use of the different template in all experiments (for example Fig. 1e, 1h, 3d, 3h, 4d, 5c, 8b, 8c and others). The authors only provide averages and standard errors for some experiments in Supplementary Table 3, which are not sufficient. Statistical analyses and p values should be reported in the main figures in order not to mislead the reader.

(5) I previously commented on the slopes of the kinetics of transcription reported in Fig. 1d in which the use of the control template seemed to display higher rate of transcription, which is contrasting with the quantification reported in Fig. 1e. The authors argue that this was only true for one replicate and that their quantification is based on 5 replicates. Nevertheless the standard errors associated to these measurements are less than 0.01 units (Supplementary Table 3, Table 3.1), which support a high reproducibility in their assay. This observation questions again the way the authors analysed their data. Reporting the curves associated to each replicate would help the reader to assess the relevance of the data.

POINT BY POINT RESPONSE (2nd round)

We thank the reviewers for taking the extra time to point out remaining issues to clarify and analyze further. We have taken the past 3 weeks to run new experiments, rerun tests to further confirm our results, reanalyzed and replotted some results as recommended by the reviewers. Please see below how we resolved all the issues raised by the reviewers. Our answers are colored blue, headed by an arrow.

Reviewer #2 (Remarks to the Author):

The authors have done a respectable amount of additional work to address reviewer comments, including acquiring additional data and new analysis. They have also made the requested changes in the manuscript to clarify and explain the issues I raised. I am in general satisfied with the new version of the manuscript and explanations of the authors.

There are several minor issues that are either not clear to me or I thought the explanations provided by the authors or their interpretations were speculative. I will list these issues below as these reviews are also made available to the readers. But I do not think these should prevent or delay publication of the manuscript. I leave it to the discretion of the authors whether they consider these worthwhile to address in the final manuscript.

1- The 0.9 FRET peak in Figure 7 is interpreted as the state that has both an R-loop and a G4. However, the only population that we know with certainty to have such high FRET value is the G4 state. I understand that G4 comes after R-loop formation but persists even after the R-loop is removed. So, I believe this population consists both G4 with R-loop and G4 without R-loop. This perspective is also supported by the reduction in the 0.9 FRET peak upon RNaseH treatment (Fig. 7c). The reduction in the 0.9 FRET peak might be due to the loss of G4 with R-loop population and what remains is the G4 without R-loop population. Considering the 0.9 FRET to be R-loop with G4 makes it more convenient to correlate with the data on the other constructs, but I don't think we can conclude that all 0.9 FRET states have an R-loop.

7c

➤ We acknowledge our mistake of presenting the data without proper normalization. Here is the corrected version of 7c, in which each FRET histogram is normalized against the total area under the curve. The high FRET peak is quantified as fraction by multiple-Gaussian fitting. The fitting result shows high FRET peak mostly remains the same after RNase H digestion, indicating G4 structure still remains. We replaced the old Figure 7c with this new version.

2- I did not find the reasoning and evidence provided by the successive R-loop formation to be convincing, especially when these events are separated by ~ 10 s. I also did not find the explanation for why having an R-loop in place facilitates a new RNAP to start a new transcription event. After all, RNAP or the newly synthesized RNA still has to replace the old RNA that is within the R-loop. This might very well be what is happening, but it remains mechanistically vague for me. Having an R-loop from the previous transcription event might possibly help as it would prevent the template and non-template DNA strands from hybridizing, and this might facilitate the next transcription event.

- We completely agree with the reviewer's interpretation i.e RNAP replaces the old RNA that is within the R-loop. Indeed, we included the following sentences to explain this point.

"Our results indicate that the R-loop induces continuous transcription activity despite the high thermal stability. One possibility is that the R-loop presents itself as a pre-open state for RNAP to access the template strand with a lower energetic barrier than the duplex DNA. In this way the R-loop can serve as the structural conduit for increased RNA production (Fig. 1, 5d and 8b)."

3- The authors have explained the lower FRET value in R-loop state compared to dsDNA (Fig. 5d) with fluorophore dipole orientation. This is very speculative since no measurement has been performed to probe the dipole orientation in either dsDNA or the R-loop state. For all practical purposes, the orientation factor (K^2) might be greater in the R-loop state compared to the dsDNA state or vice versa. The Iqbal et al. (PNAS 2008) paper which is cited in the response is not a representative paper for the context of this study since in that paper the ends of short duplexes were labeled whereas in this study the fluorophores are internally labeled. It is generally accepted that when placed at the end of a duplex, fluorophores stack on the last base, and are restricted in terms of the orientations they can assume. The situation is quite different when fluorophores are labeled internally. I believe the difference in FRET between the R-loop and the dsDNA is due to displacement of the non-template strand, which results in lower FRET state than the R-loop state.

- We thank the reviewer for providing the detailed comment about the dipole orientation effect. We agree with the reviewer that we cannot probe the dipole orientation in the construct we used. We do not cite the PNAS 2008 paper in the manuscript. We agree with the reviewer that the difference in FRET states between the R-loop and dsDNA is due to the displacement of NT. One additional contribution may arise from the transcription bubble which may further increase the physical distance between the two dyes. This is documented in Supplementary figure 3a and 8a shown below.

S3 a

S8 a

4- I noticed a couple typos in Figure 3 caption: “300 trances” and “f, As in shown e”

➤ Typos are corrected. Thank you!

Reviewer #3 (Remarks to the Author):

In the revised version of their manuscript, Lee et al. only partially addressed the comments/issues raised by the reviewers. While the work reported herein is of potential interest, the current manuscript is not ready for publication in Nature Communications or any other journals. The main limitation of the current manuscript lies in the absence of clear evidences demonstrating that G-quadruplex (G4) structure formation in a non-template DNA is necessary for stimulating transcription. The authors did not demonstrate that the observed transcription stimulation is due to the G4 structure in the non-template strand rather than the guanine-rich sequence in the nascent RNA. Because this is the main conclusion of the manuscript and the basis of the title of the manuscript, clear evidences need to be reported. Here are some comments on the revised manuscript:

Main points:

(1) In the most recent report from the Hanawalt lab (NAR 2017), the authors report that the G-richness of a sequence rather than its ability to fold into a G4 structure affect both R-loop formation and transcription. Moreover the authors report that R-loop formation inhibits rather than stimulates transcription. The experimental setup used by the Hanawalt lab only differs from the current study by the position of the quadruplex. Studying the positional effect of the G4-forming sequences is then crucial for understanding the contribution of G4 and R-loop to transcription regulation. Without this information, the current manuscript will confuse, or even worse mislead, the potential readers when trying to understand the role of guanine-rich sequences in transcription regulation. From the answers of the authors to my comments, it seems that they already performed some experiments to investigate this point. While they agree that the position of the G4 affects the outcome of the experiments, their answer remains elusive and they do not provide any experimental data. This is detrimental to their manuscript and these data should be presented.

- We are sorry that we did not provide the position dependent data in the last revision. Here is the result on the transcription of cMyc-NT located at three different distances from the promoter; 9, 16 and 41 bp (S. figure 10a). This result clearly indicates that the position of PQS i.e the distance between the promoter and first G of PQS influences the R-loop and transcriptional output (S. figure 10b, c). For your information, the position used in Hanawalt's paper was 9 base pairs from the promoter whereas we used 16 bp and 41bp positions in our assays.
- R-loop fraction was quantified by Cy5 signal within the same lane. Since the same Cy5 primer was used to PCR-purify all three DNAs, the relative Cy5 intensity indicates the relative DNA concentrations for both the dsDNA and R-looped DNA (up shifted). RNA production was quantified by green signal (stained with SybrGreen II) and normalized by the total DNA signal (Cy5) to calculate the RNA production per the same amount of DNA.
- As shown below, R-loop formation at 9 and 16bp positions are higher than that of 41bp position (S. figure 10b) while the transcription is lowest at 9 bp and substantially higher at 16 and 41 bp (S. figure 10c). Therefore, our data is consistent with that of Hanawalt's result (NAR, 2017) in which they used 9 bp position which reduced the transcription. We do not fully understand the reason behind this reduction in transcription, but we think it may have to do with its proximity to transcription initiation

site in which transcription bubble formation within the R-loop context may be rate limiting i.e slowing down transcription whereas 16 and 41 bp represent positions of actively elongating RNAP in which the R-loop results in enhanced transcription. Now we include this data (S. figure 10) to avoid the source of confusion.

10a

10b

10c

(2) The authors have now including experiments studying the impact of G4 stabilisation, using KCl or NMM, on transcription. Nevertheless the results presented in the “bubble” graphs (Fig. 8b and 8c) do not seem to agree with the raw data (EMSA presented in Supplementary Fig. S9). It is unclear how the transcription efficiencies from the different templates and conditions have been quantified. One can see from the gel reported in Supplementary Fig. S9a that the use of the control template leads to the

highest transcription efficiencies and that the addition of KCl or NMM do not affect transcriptional outputs.

- We thank the reviewer for pointing this out. We have now added the quantification and normalization method in the method section. To illustrate clearer, we also plot the result as bar graphs (new fig. 8b-d) shown below. The new plot demonstrates the clear difference in transcription induced by different buffer conditions i.e. less stable G4 (no salt, LiCl vs. highly stable G4 conditions (KCl, NMM) induce low vs. high transcription activity, respectively. We replaced the previous bubble figure by this more quantitative plot.

Moreover it seems that an R-loop can also be formed when performing the quadruplex on the template strand. All these observations are in clear contradiction with the data reported in the manuscript (Fig. 8b).

- This is because the pre-folding is never 100% although we freshly re-anneal the DNA for every experiment. Therefore, we can still observe small amount of R-loop formation by linear dsDNA of template construct. However, we show in a direct way that the pre-formed G4 in template does not form R-loop by smFRET 2 experiment (see below). As shown, both preformed G4 in template and non-template are high FRET at the beginning, but only non-template shifts to 0.1 state after addition of RNAP and NTP. We also check the time trace to confirm there is no transition for the template construct.

S5a

S5c

If the author's claims are real, they need to report a thorough quantification of transcriptional outputs using replicated data together with a statistical analysis of the differences in transcription. As noted by all reviewers, the use of PQS-T does not show a significant difference from the control template. Taken together these observations challenge the main conclusions of the article.

- The statistical analysis was added in the supplementary table 3, denoted as *, **, *** referring to p value below 0.1, 0.5 and 0.01, respectively (see below).
- We understand the reviewer's considerations about PQS-T, which does not show a strong difference from the control. Indeed, we do not observe a large difference between the two. Nevertheless, we do observe a significance between template and non-template, which is the main focus of our study.

(3) The authors provide, in the revised version of their manuscript, experiments performed with ITP in order to destabilise the R-loop without affecting G4 formation. However the reported data are either conflicting or incomplete. From the gel reported in Supplementary Fig. 9b, one can see that the use ITP globally inhibit the transcription of all templates as RNA products can barely been seen. This suggests that, under the condition of the assay, ITP is a general inhibitor of transcription. The method for RNA quantification should be reported appropriately, replicated and the relevance of any difference statistically analysed. The authors nevertheless reports some sort of quantification on Fig. 8c, but the data associated with the different conditions (LiCl, KCl, NMM, preformed) are missing. These data are the most informative in order to conclude on the role of the G4 motif in the absence of an R-loop. If no differences are observed, one may conclude that the G4 motif does not contribute to transcription regulation but its formation is merely the consequence of the formation of the R-loop.

- Here we have provide both NTP and ITP experiments. We also updated the figure by including a newly performed gel image and analysis which contains much cleaner RNA bands in ITP

experiment (see S. figure 9b, below). Due to the lower transcript in ITP condition, we stained the gel longer to make the RNA bands clearer. We have repeated this experiments several times by staining RNA for 5 to 30 minutes and obtained similar relative RNA production plotted in S. figure 9e shown below.

- The quantification method is the same as stated above. However, due to the difference band intensities of RNA, the RNA production was normalized only within one gel. Every repeated experiment was normalized separately to avoid the random errors among each gel.

S9b 1 mM ITP mix

8d

- It is clear that in ITP condition, there is no difference among control, template and non-template PQS in varying buffer conditions. This in part suggests R-loop rather than G4 plays a more important role in regulating transcription. We do not conclude (also not written in manuscript) that G4 motif itself contributes to transcriptional regulation independently. The main driving force (as we state in manuscript) is the R-loop. The additional G4 formation, however stabilizes the R-loop structure and thereby stimulating higher transcription activity.

(4) It is crucial that the authors carefully assess the statistical differences associated to the use of the different template in all experiments (for example Fig. 1e, 1h, 3d, 3h, 4d, 5c, 8b, 8c and others). The authors only provide averages and standard errors for some experiments in Supplementary Table 3, which are not sufficient. Statistical analyses and p values should be reported in the main figures in order not to mislead the reader.

- The statistical analyses (p-value) were updated in the Supplementary table 3. The figures were also updated to indicate the significance as shown below, except 3d, 3h and S2e, which have no significant differences.
- For figure 1e, we only emphasize the difference between template and non-template, where each p-value was analyzed based on the same PQS with either orientations.

(5) I previously commented on the slopes of the kinetics of transcription reported in Fig. 1d in which the use of the control template seemed to display higher rate of transcription, which is contrasting with the quantification reported in Fig. 1e. The authors argue that this was only true for one replicate and that their quantification is based on 5 replicates. Nevertheless the standard errors associated to these measurements are less than 0.01 units (Supplementary Table 3, Table 3.1), which support a high reproducibility in their assay. This observation questions again the way the authors analysed their data.

Reporting the curves associated to each replicate would help the reader to assess the relevance of the data.

- To clearly address the difference between template and non-template, we select four representative PQS (from strong to weak G4) and plot the result obtained in 60 min window. We used first 20 min (linear portion part) to calculate the rate of Cy3 increase plotted in Figure 1e above.

- In last revision, the raw data of each PQS was fitted by linear equation and normalized to control data tested in the same set of experiment. The overall standard errors were calculated by percentage error from each fitted result, which is based on chi-square test of the fitting curve. However, as reviewer points out, we also found this makes the standard error too narrow to reflect the real situation. Therefore, we have updated the standard error by calculating from the original data. Here shows the updated figure and table, where we provide experimental data based on six independent experiments. The statistical analysis was calculated by t-test. To simplify the table, we only compare the difference between each PQS to control and the difference between template and non-template.

Supplementary Table 3-1

Name	Repeated experiments						Average Rate	Standard error	p-value	
									Control	Template
Control	1.000	1.000	1.000	1.000	1.000	1.000	1.000	0.030		
111-T	0.806	0.894	0.997	0.957	0.942	0.982	0.930	0.026	< 0.05	
cMyc-T	1.016	1.193	1.148	0.954	0.992	1.105	1.068	0.035	< 0.1	
144-T	0.882	0.996	0.987	0.994	0.949	0.946	0.959	0.016	< 0.05	
222-T	0.807	0.959	0.732	0.815	0.854	0.805	0.829	0.028	<0.01	
199-T	0.982	1.014	1.099	0.965	0.941	0.975	0.996	0.021	0.842	
333-T	0.906	1.072	1.227	0.985	0.975	0.915	1.013	0.045	0.751	
TTA-T	0.802	1.000	1.047	1.040	1.031	0.691	0.935	0.056	0.237	
555-T	0.850	0.985	1.032	1.107	1.058	0.954	0.998	0.034	0.940	
AAA-T	0.920	0.911	1.044	0.975	0.943	0.921	0.952	0.019	< 0.05	
111-NT	1.093	0.961	1.155	1.211	1.113	1.541	1.179	0.073	< 0.05	< 0.01
cMyc-NT	1.218	1.252	1.237	1.176	1.553	1.754	1.365	0.087	< 0.01	< 0.01
144-NT	1.198	1.184	1.182	1.302	1.251	1.175	1.216	0.019	< 0.01	< 0.01
222-NT	1.245	1.386	1.534	1.254	1.246	1.155	1.303	0.050	< 0.01	< 0.01
199-NT	0.970	1.195	1.259	1.251	1.974	1.138	1.298	0.130	< 0.05	< 0.05
333-NT	1.229	1.027	1.352	1.391	1.418	1.312	1.288	0.054	< 0.01	< 0.01
TTA-NT	1.068	1.228	1.201	1.155	1.154	1.154	1.160	0.020	< 0.01	< 0.01
555-NT	1.156	1.072	1.270	1.282	1.230	1.585	1.266	0.065	< 0.01	< 0.01
AAA-NT	1.218	1.001	1.186	1.311	1.195	1.254	1.194	0.039	< 0.01	< 0.01

* Standard error is presented as 95% confidence limit.

* p-value was reported by t-test.

REVIEWERS' COMMENTS:

Reviewer #2 (Remarks to the Author):

The authors have satisfactorily responded to my comments and have made the changes I suggested.

Reviewer #3 (Remarks to the Author):

The authors have now addressed my concerns and comments. The latest version of the manuscript by Lee et al. will be of great interest for the readership of Nat Commun.